# MoMaGen: Generating Demonstrations under Soft and Hard Constraints for Multi-Step Bimanual Mobile Manipulation

**Chengshu Li[1*], Mengdi Xu[1*], Arpit Bahety[2*], Hang Yin[1*], Yunfan Jiang[1], Huang Huang[1], Josiah Wong[1], Sujay Garlanka[1], Cem Gokmen[1], Ruohan Zhang[1], Weiyu Liu[1], Jiajun Wu[1], Roberto Martín-Martín[2,3], Li Fei-Fei[1]**
[1]Stanford University, [2]The University of Texas at Austin, [3]Amazon. [*]Equal Contribution

## Abstract

Imitation learning from large-scale, diverse human demonstrations has been shown to be effective for training robots, but collecting such data is costly and time-consuming. This challenge intensifies for multi-step bimanual mobile manipulation, where humans must teleoperate both the mobile base and two high-DoF arms. Prior X-Gen works have developed automated data generation frameworks for static (bimanual) manipulation tasks, augmenting a few human demos in simulation with novel scene configurations to synthesize large-scale datasets. However, prior works fall short for bimanual mobile manipulation tasks for two major reasons: 1) a mobile base introduces the problem of how to place the robot base to enable downstream manipulation (reachability) and 2) an active camera introduces the problem of how to position the camera to generate data for a visuomotor policy (visibility). To address these challenges, MoMaGen formulates data generation as a constrained optimization problem that satisfies hard constraints (e.g., reachability) while balancing soft constraints (e.g., visibility while navigation). This formulation generalizes across most existing automated data generation approaches and offers a principled foundation for developing future methods. We evaluate on four multi-step bimanual mobile manipulation tasks and find that MoMaGen enables the generation of much more diverse datasets than previous methods. As a result of the dataset diversity, we also show that the data generated by MoMaGen can be used to train successful imitation learning policies using a single source demo. Furthermore, the trained policy can be fine-tuned with a very small amount of real-world data (40 demos) to be succesfully deployed on real robotic hardware. More details are on our project page: `momagen.github.io`.

## 1 Introduction

Learning from human demonstrations is a powerful paradigm for teaching robots complex manipulation skills. A common approach for collecting such data is teleoperation, where a human directly controls the robot to demonstrate desired behaviors. When scaled up, teleoperated data has enabled training visuomotor policies with impressive generalization and success in challenging manipulation tasks (Brohan et al., 2022b; O'Neill et al., 2024; Khazatsky et al., 2024; Black et al., 2024; Intelligence et al., 2025). However, this data collection process remains expensive and time-consuming, especially for tasks that require high-quality demonstrations to ensure effective policy learning.

Recently, collecting a small amount of human teleoperation data and then synthesizing additional data in simulation has become a popular approach to scale up data collection (Mandlekar et al., 2023b; Garrett et al., 2024; Jiang et al., 2025b; Xue et al., 2025; Yang et al., 2025). Compared to offline data augmentation techniques such as those based on image augmentation (Laskin et al., 2020; Yarats et al., 2021; Pitis et al., 2020; Yu et al., 2023; Chen et al., 2023), this approach can autonomously generate new behaviorally diverse data for the same task. This process enlarges the support and convergence region of the policies and reduces the teacher-student distribution mismatch (Ross et al., 2011) through new generated experiences validated in simulation to ensure quality. Notably, the X-Gen family of techniques (Mandlekar et al., 2023b; Garrett et al., 2024; Jiang et al., 2025b; Xue et al., 2025; Yang et al., 2025) leverages simulation and augmentation based on a small number of human demonstrations that are used as seeds to generate multiple new variations automatically. While they have shown success in simple table-top manipulation tasks, a significant standing challenge for

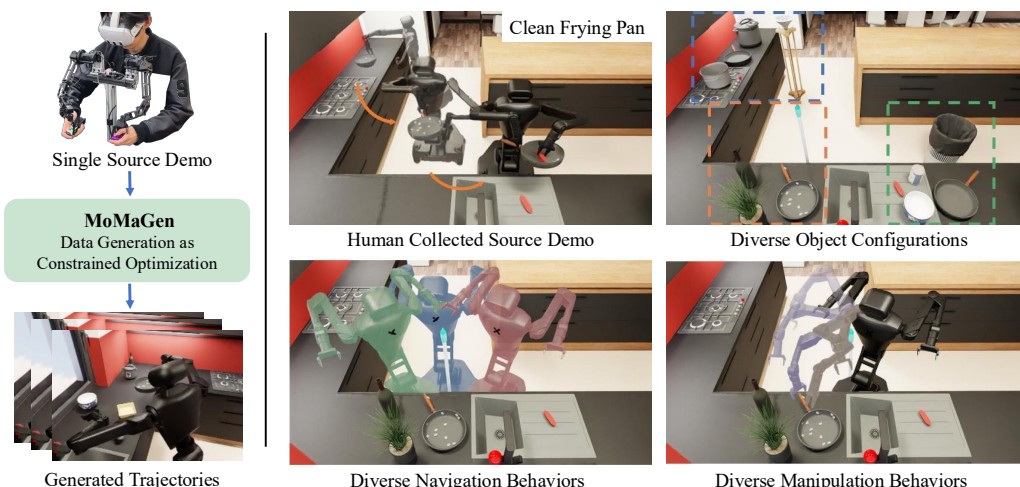

Figure 1: (left) MOMAGEN uses a single human-collected demonstration to generate a large set of demonstrations, formulating data generation as a constrained optimization problem. (top-left) shows a human-collected demo for cleaning a pan with a scrub. (top-right) shows three novel object configurations with aggressive object pose randomization and additional distractors/obstacles. MOMAGEN can generate novel trajectories in these diverse scenarios. (bottom-left) shows three robot base poses and (bottom-right) shows two arm trajectories for picking up the pan.

X-Gen methods is to extend the benefits of generalizable data generation to real-world tasks with more complex robot embodiments such as mobile manipulators.

Solving real-world tasks, such as everyday household activities, often requires a mobile manipulator with whole-body control capabilities to coordinate stable and accurate navigation with end-effector manipulability, often with two arms (Jiang et al., 2025a; Li et al., 2020; Fu et al., 2024b). Teleoperation data collection becomes significantly challenging for high-degrees-of-freedom whole-body control since controlling the base and two arms is a severe overload on the human operators (Dass et al., 2024; Ishiguro et al., 2020; Fu et al., 2024b) (see Fig. 1, *left*). Augmenting a few (expensive) demonstrations becomes thus critical, but previous methods of the X-Gen family fall short in this domain due to two major reasons: First, mobile manipulation introduces the problem of **object reachability**. For novel object arrangements, naive replay of the navigation segments of the human collected demonstrations easily leads to robot configurations that render subsequent manipulation infeasible. Second, having a mobile base, and thus a movable camera, exacerbates the problems of partial observability. Concretely, when training visuomotor policies for mobile manipulation, a naive augmentation of demonstrations leads to severe problems in **object visibility**: the task-relevant objects may move out of the field of view, making it hard for the policy to make optimal decisions based on the images from onboard sensors. Naive motion planning (Garrett et al., 2024) or replay (Mandlekar et al., 2023a; Jiang et al., 2025b) is insufficient to ensure either reachability or visibility of the task-relevant objects.

To address these challenges, we propose MOMAGEN, a general data generation method for bimanual mobile manipulation. It formulates data generation as a constrained optimization problem with hard constraints (e.g., reachability, visibility during manipulation) and soft constraints (e.g., visibility while navigation). Significantly, we realized that previous methods of the X-Gen family can be interpreted with the same unified framework, except using different (and insufficient) hard and soft constraints for data generation. We evaluate MOMAGEN on four multi-step bimanual mobile manipulation tasks and find that it generates substantially more diverse datasets than prior methods. Leveraging this diversity, we show that the synthetic data can train effective imitation learning policies from just a single demonstration. Moreover, these policies can be fine-tuned with a small amount of real-world data (40 demonstrations) to achieve successful deployment on physical robotic hardware.

## 2 RELATED WORKS

**Data Acquisition for Robot Learning.** Collecting large-scale human teleoperation data for robot learning incurs considerable costs. Scaling up data collection requires a large number of human operators over extended periods of time (Brohan et al., 2022b; O'Neill et al., 2024; Khazatsky et al., 2024). Offline data augmentation techniques can boost data quantity and diversity by perturbing

| Methods | Bimanual | Mobile | Obstacles | Base Random. | Active Perception | Hard Constraints | Soft Constraints |
|---|---|---|---|---|---|---|---|
| MimicGen (Mandlekar et al., 2023b) | ✗ | ✓ | ✗ | ✗ | ✗ | Succ | N/A |
| SkillMimicGen (Garrett et al., 2024) | ✗ | ✗ | ✓ | ✗ | ✗ | Succ, Kin, C-Free | N/A |
| DexMimicGen (Jiang et al., 2025b) | ✓ | ✗ | ✗ | ✗ | ✗ | Succ, Temp | N/A |
| DemoGen (Xue et al., 2025) | ✗ | ✗ | ✓ | ✗ | ✗ | Kin, C-Free | N/A |
| PhysicsGen (Yang et al., 2025) | ✓ | ✗ | ✗ | ✗ | ✗ | Kin, C-Free, Dyn | Trac |
| MoMaGen (Ours) | ✓ | ✓ | ✓ | ✓ | ✓ | Succ, Kin, C-Free, Temp, Vis | Vis, Ret |

Table 1: Comparison of different automated data generation methods and the constraints they enforce. "Succ": task success; "Kin": kinematic feasibility; "C-Free": collision-free execution; "Temp": temporal constraints for bimanual coordination; "Dyn": system dynamics; "Trac": target trajectory tracking; "Vis": visibility of task-relevant objects in the robot's camera view; "Ret": retraction of robot torso and arm to a compact configuration before navigation.

existing trajectories (Mandlekar et al., 2023b; Garrett et al., 2024), or leveraging image augmentation techniques and generative models (Laskin et al., 2020; Yarats et al., 2021; Pitis et al., 2020; Yu et al., 2023; Chen et al., 2023). However, the augmented data may not always be executable by real robots. A promising alternative is to leverage automated data generation and validation in simulation. Fully automated approaches include trial-and-error (Levine et al., 2018; Pinto & Gupta, 2016; Yu et al., 2016; Dasari et al., 2019) and pre-programmed (e.g., scripted) experts (James et al., 2020; Gu et al., 2023; Jiang et al., 2022; Wang et al., 2023), which are yet to be proven effective for multi-step complex tasks with contact-rich interaction. X-Gen (Mandlekar et al., 2023b; Garrett et al., 2024; Jiang et al., 2025b; Xue et al., 2025; Yang et al., 2025) represents a hybrid approach that uses a handful of human demonstrations as seeds to generate many new variations, augmenting the data by a factor of $25\times$ to $350\times$ (Mandlekar et al., 2023b; Jiang et al., 2025b), while ensuring synthesized data is valid in simulation. A comparison between prior X-Gen works and our work be found in Table 1. Our work signifies an important step toward a more generalizable data generation framework for challenging mobile manipulation tasks, which have never been tackled before.

**Imitation Learning for Mobile Manipulation.** Early successes in robot imitation learning mostly focused on fixed-based arms, but many real-world tasks require a mobile manipulator that can both navigate and manipulate. Such robots need to effectively navigate in an environment to position themselves for downstream manipulation. Collecting teleoperation data for mobile manipulation is significantly more costly: operators must simultaneously control the robot base and arms (Wong et al., 2022; Fu et al., 2024b; Shaw et al., 2024; Dass et al., 2024; Jiang et al., 2025a), calling for automated data generation methods for better scaling. On the algorithmic side, imitation learning methods started handle the complexities of mobile manipulation tasks, employing behavior cloning (Brohan et al., 2022a; Fu et al., 2024b;a; Cheng et al., 2024; Wu et al., 2024a; Yang et al., 2024; Li et al., 2024b; Ze et al., 2024; He et al., 2024) and large pretrained models (Ichter et al., 2022; Wu et al., 2023; Xu et al., 2023; Stone et al., 2023; Jiang et al., 2024b; Shah et al., 2024; Wu et al., 2024b).

## 3 PROBLEM FORMULATION: AUTOMATED DATA GENERATION AS CONSTRAINED OPTIMIZATION

We formulate automated demonstration data generation as a constrained optimization problem, and provide a unified framework that incorporates existing approaches (see Table 1). This optimization problem includes both hard and soft constraints. The former must be strictly satisfied (e.g., task success, convergence to target end-effector poses at key frames, and collision avoidance). The latter capture desirable properties (e.g., shorter trajectory and reduced jerkiness). Generating valid data requires strictly satisfying hard constraints while minimizing the costs associated with soft constraints.

Each task is modeled as a Markov Decision Process (MDP) with state space $\mathcal{S}$ and action space $\mathcal{A}$. Given a set of source demonstrations $\mathcal{D}_{src} = \{d^j = (s_0^j, a_0^j, \ldots, s_{T_{src}}^j)\}_{j=0}^{N_{src}}$, where $N_{src}$ is the number of source demonstrations, $T_{src}$ is the trajectory length, and $s_0 \sim D$ is the initial state from distribution $D$. We aim to generate a new set of successful demonstrations $\mathcal{D} = \{d\}^{N_{gen}}$ given the source demonstrations $\mathcal{D}_{src}$ and a set of constraints $\{\mathcal{G}_i\}$. With the generated demonstrations, we can train Behavioral Cloning (Pomerleau, 1988) policies $\pi_\theta$ using $\arg\min_\theta \mathbb{E}_{(s,a)\sim\mathcal{D}}[-\log \pi_\theta(a|s)]$.

Following prior work (Mandlekar et al., 2023b; Garrett et al., 2024; Jiang et al., 2025b), each source demonstration $d$ can be decomposed to $N$ subtasks. Each subtask contains an object trajectory $S_i(o_i), i \in [N]$, where $o_i$ is the object of interest (i.e. target object), and an end-effector trajectory $\tau_i = \{\mathbf{T}_W^{E_k}\}_{k=0}^{K_i}, i \in [N], k \in [K_i]$, where $\mathbf{T}_W^{E_k}$ is the pose of the end-effector frame $E$ with respect to the world reference frame $W$ at time $k$, and $K_i$ is the number of steps for the subtask $i$. Each

subtask can further be labeled as either a *free-space* subtask or a *contact-rich* subtask. In a free-space subtask, the goal is to move the robot base or arms in free space (e.g. to a pregrasp pose), where feasible trajectories can be sampled using motion planning subject to kinematic and collision constraints. In a contact-rich subtask, the goal is to manipulate the relevant objects through contacts (e.g. grasping, wiping). The relative poses between the end-effector frame and the target object frame for contact-rich subtasks are preserved from the source to the generated demonstrations.

Generating a demonstration can be viewed as solving the following constrained optimization problem:

$$
\arg\min_{a_{t \in [T]}} \mathcal{L}(\cdot) \quad \text{s.t.} \quad
\begin{cases}
s_{t+1} = f(s_t, a_t), & \forall\, t \in [T] \\
\mathcal{G}_{\text{kin}}(s_t, a_t) \leq 0, & \forall\, t \in [T] \\
\mathcal{G}_{\text{coll}}(s_t, a_t) \geq 0, & \forall\, t \in [T] \\
\mathcal{G}_{\text{vis}}(s_t, a_t, o_{i(t)}) \leq 0, & \forall\, t \in [T] \\
\mathbf{T}_W^{E_k} = \mathbf{T}_W^{o_i}(\mathbf{T}_W^{o_i,src})^{-1}\mathbf{T}_W^{E_k}, & \forall\, contact\ \tau_i,\ \forall\, k \in [K_i] \\
s_t \in D_{\text{success}} & \exists\, t \in [T]\ \text{(task success)}
\end{cases}
\tag{1}
$$

Here, $\mathcal{L}(\cdot)$ contains user-specified soft constraints. The function $f(s_t, a_t)$ denotes the system dynamics. The constraints $\mathcal{G}_{\text{kin}}$ encode kinematic feasibility (e.g. joint limits), $\mathcal{G}_{\text{coll}}$ encode collision avoidance, and $\mathcal{G}_{\text{vis}}$ encode visibility constraints (e.g. during manipulation).

## 4 MoMaGen

Following the proposed problem formulation in Section 3, we develop MoMaGen that solves a constrained optimization problem to generate demonstrations for bimanual mobile manipulation tasks. We first introduce the reachability and visibility constraints that are essential for bimanual mobile manipulation in Section 4.1. We then detail the data generation method in Section 4.2.

### 4.1 Constraints for Bimanual Mobile Manipulation

In our instantiation of MoMaGen, beyond the commonly used constraints discussed earlier, we introduce several key technical innovations that are crucial for generating high-quality bimanual mobile manipulation demonstrations, regarding reachability, visbility and retraction.

**Reachability as Hard Constraint.** A key distinction of mobile manipulation is the need to control the robot base for effective downstream manipulation. Prior works (Mandlekar et al., 2023a; Jiang et al., 2025b) reuse base trajectories from source demos without adaptation, which fails when randomized target objects lie outside the arm's workspace. To address this, we impose reachability as a hard constraint, ensuring sampled base poses keep all required end-effector trajectories within reach.

**Object Visibility during Manipulation as Hard Constraint.** A valid base pose must also satisfy a hard visibility constraint: task-relevant objects must remain in view. Since the generated data is meant to train visuomotor policies, we ensure each sampled pose allows the head camera to observe these objects without occlusion, using the additional camera or torso articulation when needed.

**Object Visibility during Navigation as Soft Constraint.** Maintaining task-relevant object visibility during navigation is desirable but not required, so we treat it as a soft constraint by adding a visibility cost that biases the head camera toward the target object during navigation.

**Retraction as Soft Constraint.** After manipulation, the robot retracts its torso and arms into a compact configuration, reducing its footprint and making subsequent navigation safer.

As illustrated above, the choice of constraints, particularly soft constraints, is highly dependent on the specific application or domain. In our case, we selected the aforementioned constraints because we believe they promote the generation of high-quality bimanual manipulation demonstrations that closely approximate human-level optimality for visuomotor policy training.

### 4.2 Automated Demonstration Generation for Bimanual Mobile Manipulation

In this section, we discuss in detail our framework that leverages the novel constraints introduced above to efficiently generate diverse demonstration data for bimanual mobile manipulation.

**Source Demonstration Annotation.** Each demonstration is segmented into temporally ordered subtasks, that are either single-arm motions or bimanual motions requiring synchronization at subtask boundaries. For each subtask, we annotate the target object $o_{target}$, gripper-held object $o_{held}$, timestep immediately preceding contact $t_{pregrasp}$, end timestep $t_{end}$, and retraction type $r$ to execute after manipulation. Figure 2 illustrates an annotated subtask of grasping a cup with one arm. To stress-test our method and minimize human effort, we collect and annotate only a single source demo ($N_{src}$=1).

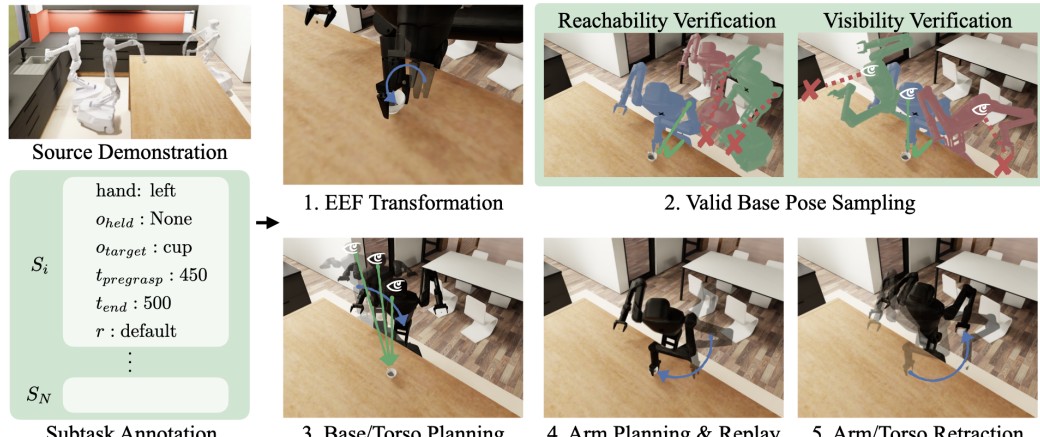

Figure 2: MoMaGen method. Given a single source demonstration, as well as annotations for object-centric subtasks for each end-effector, MoMaGen first randomizes scene configuration, and transforms the end-effector poses from the source demo to the new objects' frame of reference. For each subtask, it tries to sample a valid base pose that satisfies reachability and visibility constraints. Once found, it plans a base and torso trajectory to reach the desired base and head camera pose while trying to look at the target object during navigation. Once arrived, it plans an arm trajectory to the pregrasp pose and uses task space control for replay, before retracting back to a tucked, neutral pose.

---

**Algorithm 1** MoMaGen

**Input:** original demo, new initial state $s_0$
**Output:** generated demo

1: **for** each segment **do**
2:     Get current $\mathbf{T}^{\mathrm{base}}$, $\mathbf{T}^{\mathrm{cam}}$, $q^{\mathrm{torso}}$, $q^{\mathrm{arm}}$
3:     **if** held object not in hand **then abort**
4:     Compute transformed end-effector pose $\mathbf{T}^{\mathrm{eef}}$ using new target object pose
5:     Check visibility of target object with $\mathbf{T}^{\mathrm{cam}}$
6:     Solve IK for arm trajectory $\{q_t^{\mathrm{arm}}\}$ with current $\mathbf{T}^{\mathrm{base}}$, $\mathbf{T}^{\mathrm{cam}}$
7:     **while** not visible or no IK exists **do**
8:         Sample new base pose $\mathbf{T}^{\mathrm{base}}$
9:         Sample new camera pose $\mathbf{T}^{\mathrm{cam}}$
10:        Solve IK for arm $\{q_t^{\mathrm{arm}}\}$ and torso $\{q_t^{\mathrm{torso}}\}$ with sampled $\mathbf{T}^{\mathrm{base}}$, $\mathbf{T}^{\mathrm{cam}}$
11:        Plan motion for $\{q_t^{\mathrm{torso}}\}$ from current $\mathbf{T}^{\mathrm{base}}$ to sampled $\mathbf{T}^{\mathrm{base}}$, $\mathbf{T}^{\mathrm{cam}}$ w/ soft visibility
12:     Plan motion for $\{q_t^{\mathrm{arm}}\}$ from previous $\mathbf{T}^{\mathrm{eef}}$ to pregrasp $\mathbf{T}^{\mathrm{eef}}$
13:     Control end-effector in task space to follow transformed $\mathbf{T}^{\mathrm{eef}}$
14:     Attempt retraction

---

**Demonstration Generation.** MoMaGen generates new demonstrations for novel initial states by following Algorithm 1. For each subtask, we first verify that the robot holds the required object, aborting early if not (line 3), typically due to a failed grasp. We then compute end-effector poses for contact-rich motions by applying the transforms from the original demo (line 4). Next, we check reachability and visibility constraints for the current base and head camera configuration (lines 5-6); if satisfied, the robot proceeds to manipulation (lines 12–13). Otherwise, we sample base and head camera configurations until a valid one is found (lines 7–11), using heuristics and inverse kinematics to ensure reachability and visibility (lines 8–10). Once a valid one is reached via motion planning with soft visibility constraint (line 11), the robot executes the manipulation, moving to the pregrasp pose via motion planning and replaying the contact-rich segment in task space (lines 12–13). Finally, the robot tries to retract to a canonical or prior joint configuration (line 14). This process repeats for each subtask, with motion planning and IK handled by cuRobo (Sundaralingam et al., 2023).

**Key Novelties.** Our approach introduces four main innovations for generating mobile manipulation demonstrations. **(1) Full-body Motion**: Unlike prior work focused only on end-effector pose $\mathbf{T}^{eef}$, we jointly consider $\mathbf{T}^{eef}$, head camera pose $\mathbf{T}^{cam}$, and base pose $\mathbf{T}^{base}$. **(2) Visibility Guarantee**: We enforce object visibility before manipulation (lines 5, 9) and add a soft constraint to maintain

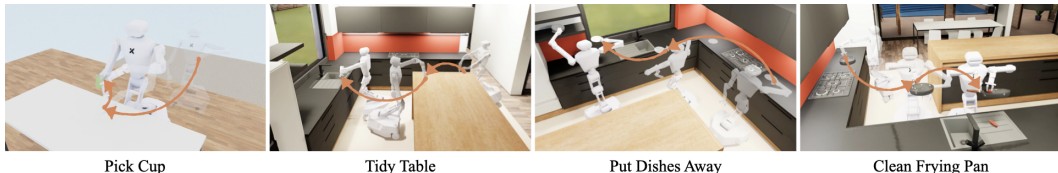

| Pick Cup | Tidy Table | Put Dishes Away | Clean Frying Pan |

Figure 3: Task visualization. Our multi-step tasks include long-range navigation, sequential and coordinated bimanual manipulation, requiring pick-and-place and contact-rich motion.

visibility during navigation (line 11). **(3) Expanded Workspace**: We sample base poses near target objects (line 8) and plan base motions across the room to fully exploit mobility (line 11). **(4) Efficient Generation**: We accelerate generation by prioritizing fast IK checks over full motion planning whenever possible for preemptive filtering, and by decomposing robot's configuration into torso/arm subspaces for efficient conditional sampling, akin to task–motion planning (Garrett et al., 2021).

## 5 EXPERIMENTS AND RESULTS

We evaluate MOMAGEN on four household tasks (Section 5.1) under three object/scene randomization schemes (Section 5.2). Compared with two baselines (Jiang et al., 2025b; Garrett et al., 2024), MOMAGEN produces much more diverse data, higher generation success rates, and substantially greater object visibility, which is crucial for visuomotor policies (Section 5.3). We further assess how these demonstrations help train imitation learning policies, with sim-to-real potentiality (Section 5.4).

### 5.1 TASK SETUP

We evaluate MOMAGEN on four household tasks that require mobile manipulation and involve various manipulation skills, including pick-and-place, contact-rich interactions and bimanual coordination, illustrated in Figure 3. Inspired by BEHAVIOR-1K, all tasks are implemented in OmniGibson (Li et al., 2023; 2024a). **Pick Cup**: navigate to a table and lift a cup. **Tidy Table**: move a cup from the countertop to the sink, requiring long-range mobile manipulation. **Put Dishes Away**: stack two plates from the countertop onto a shelf using two arms independently, requiring bimanual uncoordinated motion. **Clean Frying Pan**: use both arms to scrub a dusty pan with a brush, requiring contact-rich bimanual coordinated motion. For each task, we collect a single 1–3 minute human teleoperated demonstration, with base motion accounting for 45% of the duration.

### 5.2 DOMAIN RANDOMIZATION SCHEMES

For each task, we define three domain randomization levels of increasing difficulty. **D0**: task-relevant objects randomized within $\pm15$ cm and $\pm15°$ on the same furniture. **D1**: task-relevant objects placed anywhere on the furniture with unrestricted orientation (Figure 4a). **D2**: D1 plus additional objects on furniture (manipulation obstacles) and on the floor (navigation obstacles) (Figure 1, upper right). This scheme is significantly more aggressive than prior work (Mandlekar et al., 2023b; Garrett et al., 2024; Jiang et al., 2025b), enabled by MOMAGEN 's unique ability to generate novel base motions. More details and visualizations can be found in Sec. B.3 and Fig. 12 in the Appendix.

### 5.3 DATA GENERATION COMPARISON

We evaluate MOMAGEN for bimanual mobile manipulation under varying randomization schemes, comparing it to two baselines: SkillMimicGen (Garrett et al., 2024), which generates single-arm trajectories via motion planning and task-space control, and DexMimicGen (Jiang et al., 2025b), which targets dexterous bimanual data. Since all the evaluated tasks require substantial base movement, both baselines are extended with base trajectory replay from source demos, similar to MimicGen (Mandlekar et al., 2023a). We assess methods using three metrics: (1) data diversity (object pose and action variation), (2) generation success rate, and (3) object visibility ratio during navigation.

**How diverse are the demonstrations generated by MOMAGEN?** Figure 4a shows the variation in task-relevant object poses for the Tidy Table task under MOMAGEN (D0/D1) and SkillMimicGen (D0). The baselines succeed only in D0, as they cannot generate novel base motions. In contrast, MOMAGEN (D1) covers a much wider range of object poses, spanning the entire counter rather than a small corner. Figures 4b–c further demonstrate that MOMAGEN D1 yields substantially broader coverage of base and end-effector poses compared to D0 and the baselines. Figure 4d shows PCA projections of arm and torso joints under D0. Despite identical object pose distributions, randomizing feasible base placements allows MOMAGEN to achieve far greater action diversity. Moreover, with its motion planner, MOMAGEN can generate demonstrations in cluttered scenes with distractors or

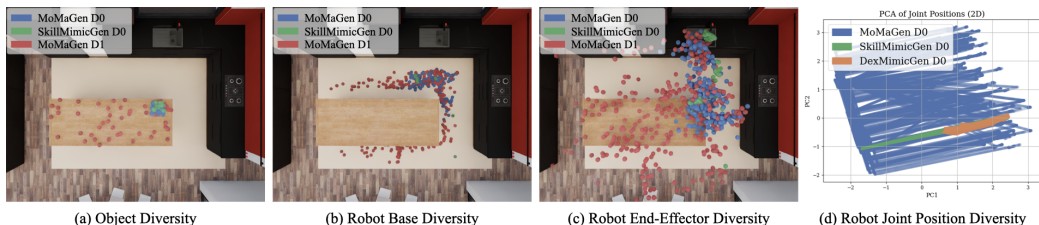

| (a) Object Diversity | (b) Robot Base Diversity | (c) Robot End-Effector Diversity | (d) Robot Joint Position Diversity |

Figure 4: Generated data diversity analysis for Tidy Table task (50 trajectories, subsampled). Given the same object randomization (D0) (a), compared to SkillMimicGen, MOMAGEN samples diverse base poses (b), and as a result, diverse end-effector poses (c) and joint positions (d). MOMAGEN is also the only method that can generate data for D1 randomization (red) for even greater diversity.

|     | Methods | Pick Cup | Tidy Table | Put Dishes Away | Clean Frying Pan |
|-----|---------|----------|------------|-----------------|------------------|
| D0  | MOMAGEN | 0.86 | **0.80** | 0.38 | **0.51** |
|     | SkillMimicGen | **1.00** | 0.69 | 0.38 | 0.40 |
|     | DexMimicGen | **1.00** | 0.72 | 0.38 | 0.35 |
|     | MOMAGEN w/o soft vis. const. | 0.88 | 0.78 | **0.50** | 0.46 |
|     | MOMAGEN w/o hard vis. const. | 0.97 | 0.59 | 0.29 | 0.24 |
|     | MOMAGEN w/o vis. const. | 0.97 | 0.74 | 0.29 | 0.36 |
| D1  | MOMAGEN | 0.60 | **0.64** | **0.34** | **0.20** |
|     | MOMAGEN w/o vis. const. | **0.66** | 0.48 | 0.23 | 0.13 |
| D2  | MOMAGEN | 0.47 | **0.22** | **0.07** | **0.16** |
|     | MOMAGEN w/o vis. const. | **0.50** | 0.16 | 0.05 | 0.12 |

Table 2: Data generation success rates comparison. For simpler tasks (Pick Cup), ablations and baselines achieve higher data gen success rates because of fewer constraints and less motion planning stochasticity. However, for more complex tasks (the other three), enforcing hard visibility constraints helps position the robot torso to a suitable configuration that facilitates downstream manipulation, leading to higher success rates. The baselines suffer from zero success rates and hence are omitted for D1/D2 because the objects are beyond the reachability of replayed base poses from the source demo.

obstacles (Figure 12), further enhancing the diversity of both robot actions and visual observations. Additional results for other tasks are shown in Figures 13, 14, and 15 in the Appendix.

**Can MOMAGEN achieve high-throughput data generation?** Table 2 shows that MOMA-GEN achieves a 63% average data generation success rate for D0 and can generate data for all tasks across all randomization levels, though throughput decreases with higher difficulty. In contrast, the baselines perform well on simple tasks like Pick Cup but fail on harder ones such as Clean Frying Pan, where adapting base motions is critical, and cannot handle D1 or D2 randomization at all.

**Can MOMAGEN generate demonstrations with high object visibility?** Because task-relevant object visibility is critical for visuo-motor policy learning (Sec. 5.4), we evaluate whether MOMAGEN indeed produces data with high object visibility. We compare it against baselines and three ablations that remove hard and/or soft visibility constraints. As shown in Table 3, MOMAGEN achieves substantially higher task-relevant object visibility (often double) relative to both baselines and ablations, especially for multi-step tasks. In the Tidy Table task (Figure 5), results further show that both hard and soft constraints are essential for maintaining high visibility during navigation.

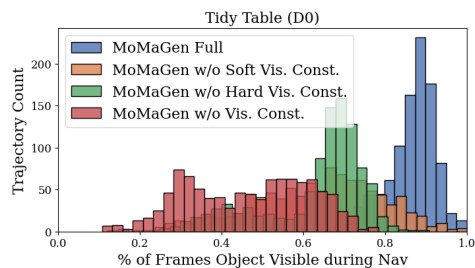

Figure 5: Object visibility analysis for MOMA-GEN and ablations. The x-axis is the % of frames where the target object is visible during navigation, and the y-axis is the trajectory count (out of 1000). MOMAGEN significantly outperforms ablations thanks to both hard and soft visibility constraints.

**Can MOMAGEN generate demos with a new robot embodiment?** We evaluate cross-embodiment data generation by using a single source demo from a Galexea R1 robot to generate Pick Cup demos on a TIAGo robot. MOMAGEN successfully transfers across platforms by planning/replaying dense end-effector trajectories in the task space, which is largely agnostic to robot-specific kinematics, demonstrating robustness and flexibility of MOMAGEN. More discussions are found in Appendix B.2.

|     | Methods | Pick Cup | Tidy Table | Put Dishes Away | Clean Frying Pan |
|-----|---------|----------|------------|-----------------|------------------|
| D0 | MOMAGEN | **1.00** | **0.86** | **0.79** | **0.69** |
|     | SkillMimicGen | **1.00** | 0.40 | 0.71 | 0.65 |
|     | DexMimicGen | **1.00** | 0.39 | 0.71 | 0.67 |
|     | MOMAGEN w/o soft vis. const. | **1.00** | 0.63 | 0.62 | 0.56 |
|     | MOMAGEN w/o hard vis. const. | 0.98 | 0.63 | 0.68 | 0.55 |
|     | MOMAGEN w/o vis. const. | 0.90 | 0.46 | 0.40 | 0.35 |
| D1 | MOMAGEN | **0.93** | **0.89** | **0.78** | **0.80** |
|     | MOMAGEN w/o vis. const. | 0.71 | 0.46 | 0.40 | 0.43 |
| D2 | MOMAGEN | **0.94** | **0.79** | **0.75** | **0.81** |
|     | MOMAGEN w/o vis. const. | 0.73 | 0.48 | 0.40 | 0.44 |

Table 3: Task-relevant object visibility comparison. Our hard and soft visibility constraints are exceedingly effective in keeping the object in view during navigation, achieving over 75% visibility even for aggressively randomized object pose (D1) and obstacles/occluders (D2). We omit baselines for D1/D2 due to zero data generation success rates.

## 5.4 POLICY LEARNING WITH GENERATED DEMONSTRATIONS

While MOMAGEN can synthesize successful trajectories for the benchmark tasks, it relies on privileged information such as ground-truth object poses and geometry. To enable real-world deployment, visuomotor policies must still be trained from onboard sensory inputs (e.g., RGB images). In this section, we investigate: (1) whether MOMAGEN generated demonstrations improve imitation learning performance over other data generation methods; (2) whether these benefits hold across different imitation learning algorithms; (3) how task-relevant object visibility influences policy training; (4) how performance scales with the number of generated demonstrations; and (5) whether a model trained on MOMAGEN data can be successfully deployed on real hardware.

**Policy Learning Setup.** We experiment with two imitation learning based methods, WB-VIMA (Jiang et al., 2025a) and $\pi_0$ (Black et al., 2024). Both methods take as input proprioceptive info and RGB images from the head camera and two wrist-mounted cameras, and outputs the target robot joint position. For WB-VIMA, we further fuse and post-process the three RGB images (with grountruth depth from the simulation) into egocentric colored point cloud, before feeding into the policy network. For WB-VIMA, we train individual single-task policies from scratch, whereas for $\pi_0$, we finetune a pre-trained $\pi_0$ model with a LoRA rank of 32. More implementation details are in Appendix C.

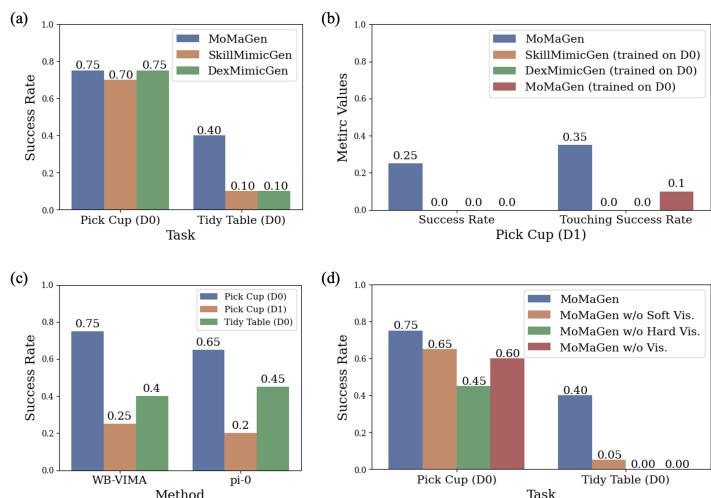

Figure 6: Comparison between MOMAGEN and other data generation methods on WB-VIMA's performances in (a) and (b), performances of WB-VIMA and $\pi_0$ trained with MOMAGEN data in (c) and visibility ablations in (d). The success rate is averaged over 20 unseen evaluation episodes. Policies trained on MOMAGEN data consistently perform better than those trained on others' data.

**How do different data generation methods impact policy performance?** We generate 1000 demonstrations with MOMAGEN for Pick Cup (D0/D1), and Tidy Table (D0), and compare against SkillMimicGen and DexMimicGen augmented with base trajectory replay. As shown in Figure 6a, MOMAGEN matches the baselines on Pick Cup (D0), where the small randomization range (0.3m × 0.3m) makes replayed navigation sufficient. On Tidy Table (D0), however, MOMAGEN significantly outperforms the baselines, which overfit to the long, nonsmooth replayed trajectories. For the more challenging Pick Cup (D1) task (1.3m × 0.8m randomization), only MOMAGEN enables WB-VIMA to achieve a 0.25 success rate, while baselines trained on D0 data fail completely (Figure 6b). The diverse base motions in MOMAGEN data also improve intermediate success like touching the cup.

**Does MoMaGen generated data benefit different imitation learning methods?** We fine-tune $\pi_0$ on MoMaGen data for Pick Cup (D0/D1) and Tidy Table (D0) using 1000 demonstrations. As shown in Figure 6c, the fine-tuned $\pi_0$ achieves success rates comparable to WB-VIMA across all three tasks. These results indicate that MoMaGen data can effectively improve the performance of diverse imitation learning approaches (either trained from scratch, or pretrained and LoRA fine-tuned).

**How does object visibility ratio affect policy performance?** In MoMaGen, the head camera is constrained to focus on the task-relevant object, aiding visual servoing during navigation and improving visibility during manipulation (Sec. 5.3). To assess the impact of these constraints on imitation learning, we conduct an ablation study with WB-VIMA on Pick Cup (D0) and Tidy Table (D0), comparing the full method with three variants: (1) without soft visibility (camera not encouraged to track the object during navigation); (2) without hard visibility (camera not enforced to observe the object during manipulation); and (3) without any visibility constraints. As shown in Figure 6d, ablations on Pick Cup (D0) achieve success rates of 0.45 to 0.65, below the 0.75 of MoMaGen. The gap is even larger on Tidy Table (D0), where ablations peak at 0.05 while MoMaGen reaches 0.40. These results demonstrate that enforcing visibility constraints during data generation substantially improves policy performance, particularly when policies rely on short history inputs.

**How is the model performance scaled with the number of MoMaGen generated demonstrations?** We extend the $\pi_0$ experiments to examine the effect of scaling MoMaGen-generated data. Models were fine-tuned with 500, 1000, and 2000 demonstrations for 50K steps and evaluated across tasks and randomization levels (Figure 7). Results show promising scaling trends, particularly under D1 randomization, where larger datasets improve coverage of both state and action spaces. These findings indicate that increasing synthetic data yields meaningful performance gains.

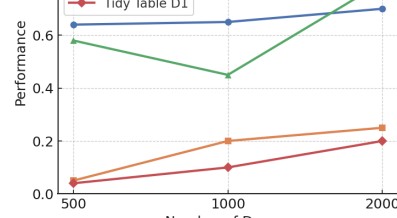

Figure 7: Data Scaling of $\pi_0$

**Can policies trained with MoMaGen' generated data be deployed in the real world?** To evaluate the real-world utility of MoMaGen, we tested the Pick Cup task using the same robot platform and a setup similar to our simulation (white table, green cup). Because zero-shot sim-to-real transfer is challenging, we collected 40 real-world demos and assessed the benefit of pretraining on 1,000 synthetic demos generated by MoMaGen. For WB-VIMA, pretraining on synthetic data followed by fine-tuning on real data achieved a 10% success rate, while training on real data alone yielded 0%. Although the overall success was low, the pretrained model consistently exhibited meaningful behavior (e.g., reaching the cup), whereas the baseline failed to progress. For $\pi_0$, the effect was stronger: the pretrained model achieved 60% success compared to 0% for the baseline. In this case, the baseline attempted grasps but failed due to poor precision, with the cup slipping from the gripper, underscoring the difficulty of learning robust policies from limited real data alone, even when starting from a strong pretrained foundation (i.e. $\pi_0$ pretrained weights). Despite the sim-to-real gap, these results show that diverse synthetic data provide a valuable prior, enabling efficient policy learning in low-data regimes. This highlights the practical utility of MoMaGen for real-world deployment (see Appendix A.1).

## 5.5 MoMaGen Failure and Computational Cost Analysis

**Failure Analysis.** We analyze the causes of unsuccessful data generation episodes by categorizing failures into base sampling, base-level planning, arm-level planning, and simulation instabilities. *Base Sampling* failures occur when no base pose is found that satisfies reachability and visibility constraints. *Base MP IK* and *Base MP TrajOpt* correspond to inverse-kinematics and trajectory-optimization failures during base motion planning. Similarly, *Arm MP IK* and *Arm MP TrajOpt* capture IK and trajectory-optimization failures during arm motion planning, while *Arm MP Other* includes additional arm-planning issues such as invalid start or goal states. *Simulation Instabilities* arise from controller inaccuracies and stochastic effects in the simulator. Fig. 8 shows that across all three domain randomizations, simulation instabilities account for a substantial share of failures (35%), which can

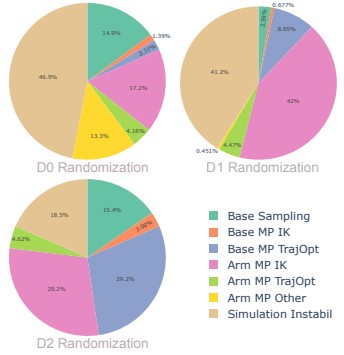

Figure 8: Failure Analysis

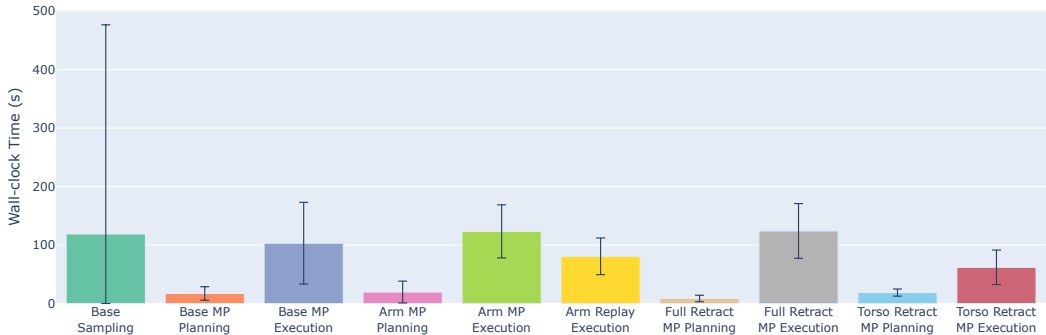

Figure 9: Compute costs (wall-clock time) for MoMaGen data generation components.

be mitigated by improvements in controller robustness and simulation fidelity. Arm-level motion planning contributes the largest proportion of planner-related failures (40% on average), exceeding base-level planning failures (26% on average). Notably, in D2 randomization, navigation-related failures (base sampling, base IK, and base TrajOpt) increase significantly due to floor obstacles in an already tight navigation space. We also analyze the distribution of failures across task-level steps within each multi-step task (Fig. 16 in Appendix).

**Compute Cost Analysis.** To better understand the efficiency of our data generation pipeline, we analyze the computational cost contribution of each component. Fig. 9 shows the wall-clock time distributions for base sampling, motion planning, and simulation execution. Simulation execution dominates total computation time, greatly exceeding the corresponding planning durations. For instance, base motion planning averages 18 seconds, whereas executing the resulting motion in simulation takes 100 seconds. Base sampling also exhibits high variance due to our current strategy of random sampling within a ring-shaped region around the target: when only a few poses are feasible, the search becomes slow, whereas abundant feasible poses lead to quick success. Future work may reduce this variance through more informed sampling methods—for example, biasing toward regions with greater free space. Fig. 10 reports the average compute cost of each component as a percentage of the total episode duration, com-

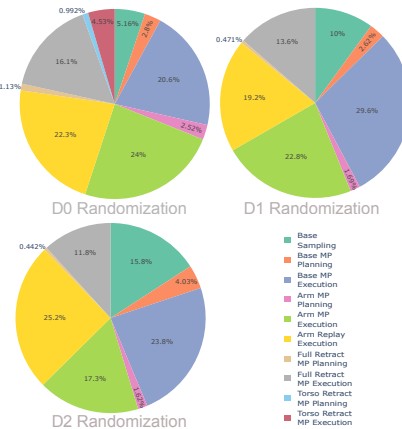

Figure 10: Compute cost of each component as a percentage of the episode

puted only over successful episodes to ensure a fair comparison across all components. Notably, base sampling becomes increasingly expensive from D0 to D2—reflecting the rise in scene complexity, where fewer feasible base poses exist and longer searches are required.

## 6 CONCLUSIONS AND LIMITATIONS

In this work, we present MoMaGen, a general data generation method for multi-step bimanual mobile manipulation using a single human-collected demonstration. MoMaGen formulates data generation as a constrained optimization problem that satisfies hard constraints while balancing soft constraints. We propose key novelties that involve reachability and visibilty constraints, and evaluate our method on four challenging bimanual mobile manipulation tasks. We showcase superior diversity and task-relevant object visibility of MoMaGen-generated data compared to those generated by baselines and ablations, which further translates to better policy learning results.

**Limitations.** We currently assume access to full scene knowledge during demonstration generation. While this is straightforward in simulation, it poses challenges in real-world scenarios. A possible solution is to incorporate vision models such as SAM2 to estimate object poses relative to the robot. Additionally, we only show data generation results with alternating phases of navigation and manipulation, although our framework is easily extensible to whole-body manipulation (e.g. opening doors) and we leave it for future work. Lastly, our approach depends on sizable GPU resources to run GPU-accelerated motion generators, which can be computationally intensive during data generation.

## REPRODUCIBILITY STATEMENT

We provide all the source code for simulation task setups, data generation, policy training, and detailed instructions for reproducing all data generation and policy learning results in our code submission (included in the supplementary materials) as well as on our website. All training hyperparameters and implementation details are documented in Appendix C to facilitate full reproducibility.

### ACKNOWLEDGMENTS

This work is in part supported by the Stanford Institute for Human-Centered AI (HAI), ONR MURI N00014-22-1-2740, ONR MURI N00014-24-1-2748, UT-CNS Catalyst Grant and Amazon Personal Robotics Group (PRG) Faculty Award.

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

## A  ADDITIONAL EXPERIMENTAL RESULTS

### A.1  SIM-TO-REAL

**Real robot setup.** We investigate whether MOMAGEN can support real-world deployment, using the Galaxea R1 mobile manipulator, the same platform used for data collection and generation in simulation. Point clouds are captured with R1's onboard ZED 2 stereo camera and an additional ZED Mini mounted above the left gripper. To mitigate depth noise, we apply the built-in AI model for point cloud refinement, followed by farthest point sampling to 4096 points, as in simulation. We evaluate on the Pick Cup (D0) task with a table of comparable height to the simulated setup and a 3D-printed green cup. The real-world configuration is shown in Figure 11a.

**Experiment setup.** Zero-shot transfer is challenging due to domain gaps, particularly for vision-based policies (Jiang et al., 2024a). In our case, simulated and real RGB images differ substantially, and real-world depth estimates are noisy. To mitigate this, we collect 50 real-world demonstrations for the Pick Cup (D0) task and evaluate whether simulation pretraining improves real-world learning.

**Sim-to-real experiments demonstrate the benefit of simulation pretraining.** For WB-VIMA, we compare models initialized from a simulation-pretrained checkpoint (1.8M steps on 1,000 synthetic demos) against random initialization. Both are fine-tuned for 35k steps on 40 real demos, with 10 held out for validation. As shown in Figure 11b, the pretrained model converges much faster, reaching a validation loss of 3.0 compared to 6.0 for the baseline. On real hardware, it achieves 10% success, while the baseline fails entirely. Despite low absolute success, the pretrained model consistently exhibits meaningful behaviors (e.g., reaching the cup), unlike the baseline, which shows no progress.

For $\pi_0$, we compare two variants initialized from pretrained weights: one further trained on simulation data before fine-tuning on 40 real demonstrations, and one fine-tuned only on real data. The simulation-pretrained model achieves 60% success, while the baseline remains at 0%. Although the baseline occasionally attempts grasps, it fails due to poor precision, highlighting the challenge of learning robust visuomotor policies from limited real data. This result shows that even with a strong foundation pretrained on 10k+ hours of robot data, it is beneficial to first adapt to the target robot embodiment and scene setup with simulation data before fine-tuning on scarce real-world data.

Overall, these results show that diverse simulation-generated data provide a strong prior, enabling efficient fine-tuning and effective sim-to-real transfer, particularly in low-data regimes. This finding demonstrates the practical utility of MOMAGEN-generated data for real-world robotics. More real-world policy rollouts can be found on our website.

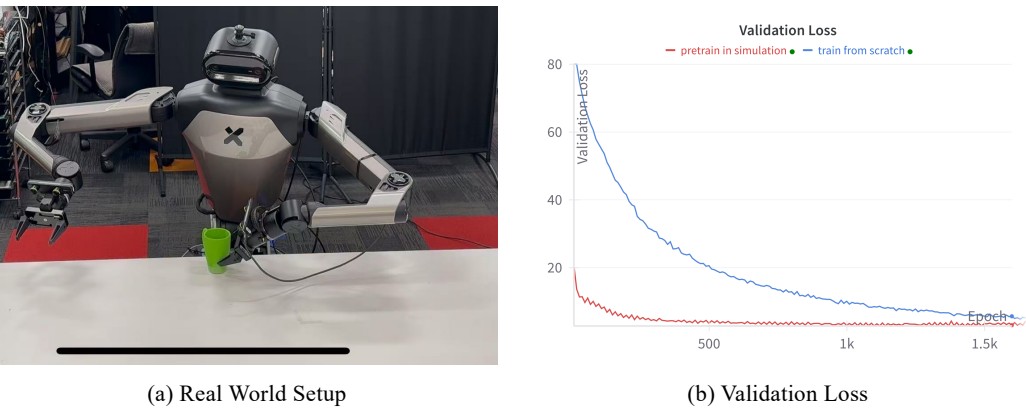

(a) Real World Setup  (b) Validation Loss

Figure 11: Real world setup for Pick Cup (a) and validation loss curve of WB-VIMA (b).

## B ADDITIONAL DATA GENERATION DETAILS

### B.1 CAN MOMAGEN GENERALIZE ACROSS DIFFERENT FAMILIES OF MOBILE MANIPULATION TASKS?

Beyond the tasks analyzed in the main paper (rigid-body pick-and-place, wiping, and stacking), we further investigate whether MOMAGEN extends to other common families of mobile manipulation behaviors. In particular, we consider three representative tasks that span different "families" of mobile manipulation: (i) articulated-object interaction, (ii) pouring and particle-transfer tasks, and (iii) whole-body control. These proof-of-concept demonstrations highlight the versatility of MOMAGEN in generating stable and collision-free trajectories across diverse tasks. Videos for all tasks are provided on our project website.

**Get Bottle from Fridge (Articulated Object Manipulation).** The robot navigates to a fridge, grasps and opens the door with one arm, and then retrieves a bottle inside. This task also demonstrates MOMAGEN's ability to manage occlusion, as the bottle is initially hidden until the fridge is opened. For the initial subtasks, the fridge is treated as the task-relevant object; once the door is open, the target switches to the bottle.

**Pour Cat Food (Pouring and Particle Interaction).** The robot grasps a tin of cat food containing rigid particles, navigates to a bowl, and performs a controlled pouring motion until all particles are transferred. This task highlights MOMAGEN's ability to generate trajectories that require coordinated container orientation, stable end-effector motion, and interaction with particle-like objects, representative of the broader family of pouring tasks.

**Push Chair (Whole-Body Control).** A chair is initially positioned outside a table. The robot navigates to the back of the chair, establishes contact, and pushes it into place underneath the table. This task requires coupling the robot's base movement with the end-effector contact direction, demonstrating that MOMAGEN can generate whole-body coordination behaviors involving sustained contact and the manipulation of large objects.

### B.2 CAN MOMAGEN ACHIEVE CROSS-EMBODIMENT DATA GENERATION?

We evaluate cross-embodiment data generation by using a single demonstration collected on a Galexea R1 robot to generate trajectories for a TIAGo robot. Although both platforms feature dual 7-DoF arms and holonomic bases, their torso designs and arm workspaces differ substantially. Our experiments show that MOMAGEN can successfully produce Pick Cup demonstrations on TIAGo using the R1 source demo. This transfer is enabled by planning and replaying dense end-effector trajectories, rather than joint-space trajectories, making the process largely agnostic to robot-specific kinematics. These results highlight the robustness and flexibility of our framework across platforms. Videos of the generated demonstrations of this new robot embodiment are available on our website.

Cross-embodiment transfer, however, has limitations. Tasks requiring operation in confined spaces may fail due to differences in gripper size, leading to unavoidable collisions. Similarly, TIAGo's bulkier arms can render some trajectories from R1 demos infeasible or even cause self-collisions. Thus, while MOMAGEN offers a proof of concept for cross-embodiment data generation, advancing more powerful approaches in this direction remains an exciting avenue for future work.

### B.3 VISUALIZATIONS OF DOMAIN RANDOMIZATION SCHEMES

Figure 12 illustrates the domain randomization schemes (D0/D1/D2) across all four tasks. Leveraging base sampling, MOMAGEN can generate successful demonstrations under more aggressive randomization (D1) than prior methods (D0). With motion planners in the loop, it further handles obstacles in both base and arm motion (D2), producing richer trajectories and more diverse visual observations.

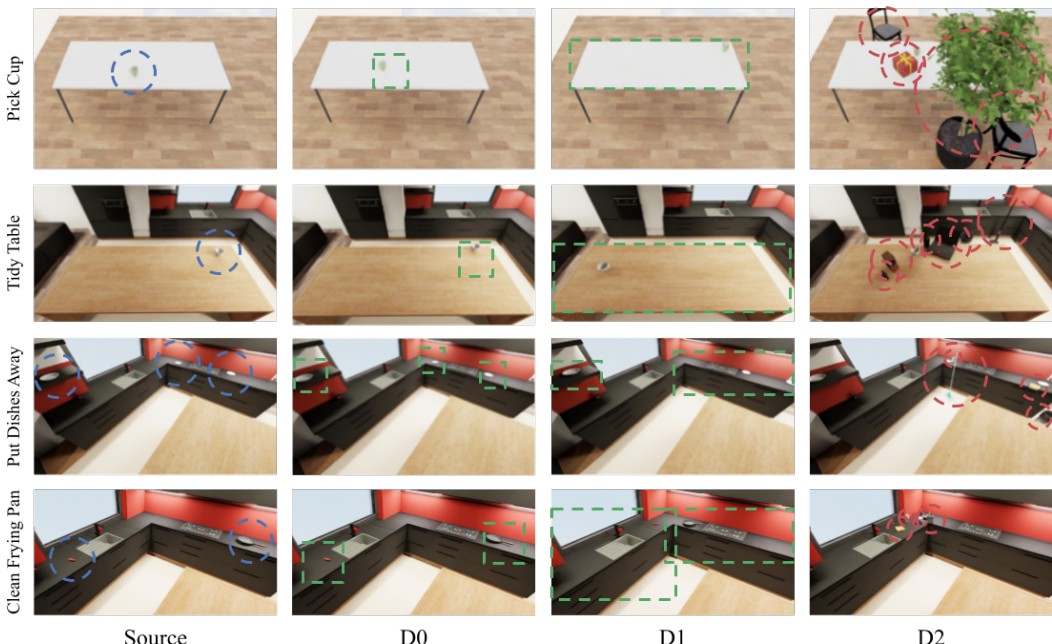

Figure 12: Visualization of Domain Randomization. Blue represents the task-relevant objects. Green represents the approximate randomization range for these objects (for D1 and D2, the range is anywhere on the same furniture). Red represents the obstacles/distractor objects.

### B.4 Visualizations of Data Diversity

Figure 13, 14, and 15 extend the visualizations in Figure 4, comparing MoMaGen to SkillMimicGen and DexMimicGen across the other three tasks. As expected, the object diversity in D1 (subfigures a) induces substantially more varied robot base and end-effector trajectories (subfigures b–c) and joint configurations (subfigure d), resulting in broader state- and action-space coverage.

### B.5 Compute Resources

The compute resource required for data generation scales with 1) the number of subtasks, 2) the length of each subtask (including the free-space subtask and contact-rich subtask), affecting motion execution time, and 3) the complexity of the scene (task-relevant objects, obstacles, etc), affecting motion planning time. It inversely scales with the data generation success rate. Each successful demonstration takes 0.1 to 1.3 GPU hours to generate, ranging from Pick Cup task to Put Dishes Away task. All data generation runs are conducted on a single NVIDIA TITAN RTX GPU.

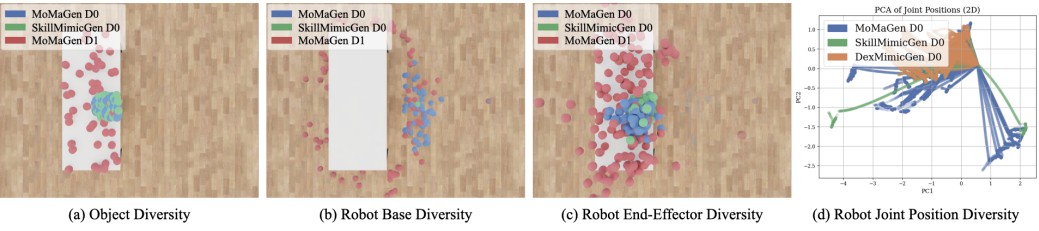

(a) Object Diversity    (b) Robot Base Diversity    (c) Robot End-Effector Diversity    (d) Robot Joint Position Diversity

Figure 13: Generated data diversity analysis for Pick Cup task (50 trajectories, subsampled).

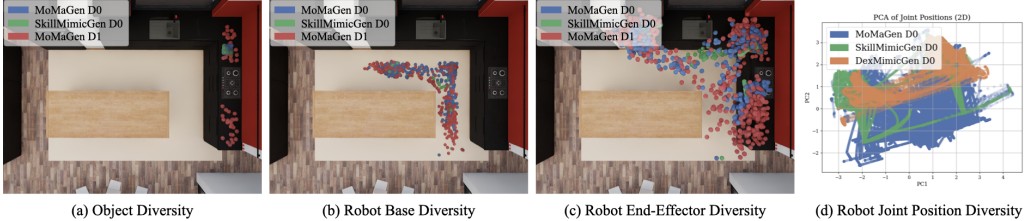

Figure 14: Generated data diversity analysis for Put Dishes Away task (50 trajectories, subsampled).

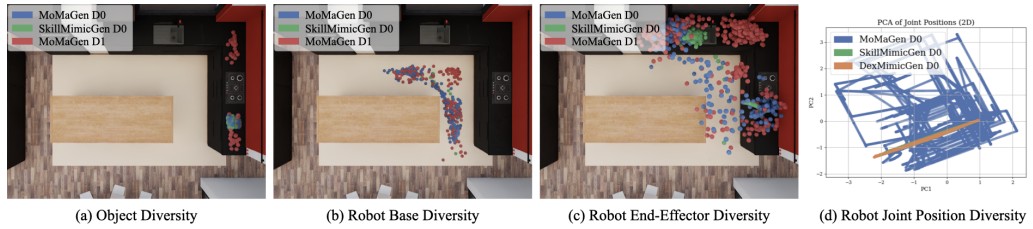

Figure 15: Generated data diversity analysis for Clean Frying Pan task (50 trajectories, subsampled).

### B.6 FAILURE STEP ANALYSIS

Fig. 16 illustrates the distribution of failures across the individual steps of each multi-step task. For example, in the put dishes away task, failures occur more frequently during the navigation and placement steps compared to the plate-picking step.

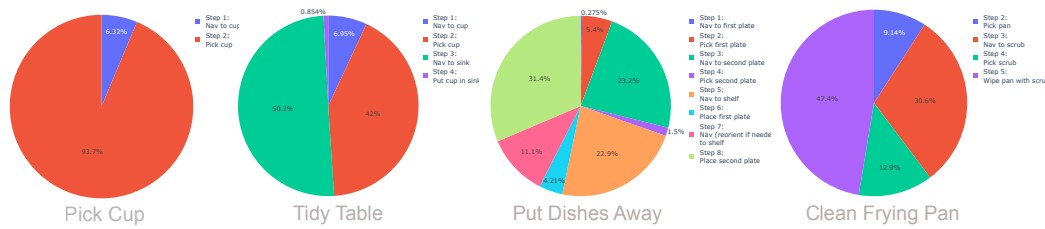

Figure 16: Distribution of failures across task-level steps within each multi-step task.

## C ADDITIONAL POLICY TRAINING DETAILS

In this section, we first describe the data cleaning process in Section C.1 and then provide additional training details for WB-VIMA (Section C.2) and $\pi_0$ (Section C.3). For both methods, 90% of the demonstrations are used for training and the remaining 10% for validation.

### C.1 DATA CLEANING

Teleoperation in simulation makes depth perception difficult, often causing operators to hesitate and leave the gripper nearly stationary before grasping or making contact. As a result, some demonstrations contain short "frozen" segments, which can degrade training, especially for imitation learning methods with limited temporal context (e.g., WB-VIMA uses a 2-step history, and $\pi_0$ relies only on the current state). To address this, we introduce a preprocessing step that removes frozen segments. Speifically, for a trajectory of length $T$, if at any timestep $i \in [0, T-5]$ the absolute joint position difference between step $i$ and $i+5$ is below $1 \times 10^{-3}$ in all dimensions, we treat the interval $[i, i+5]$ as frozen and discard it prior to policy learning.

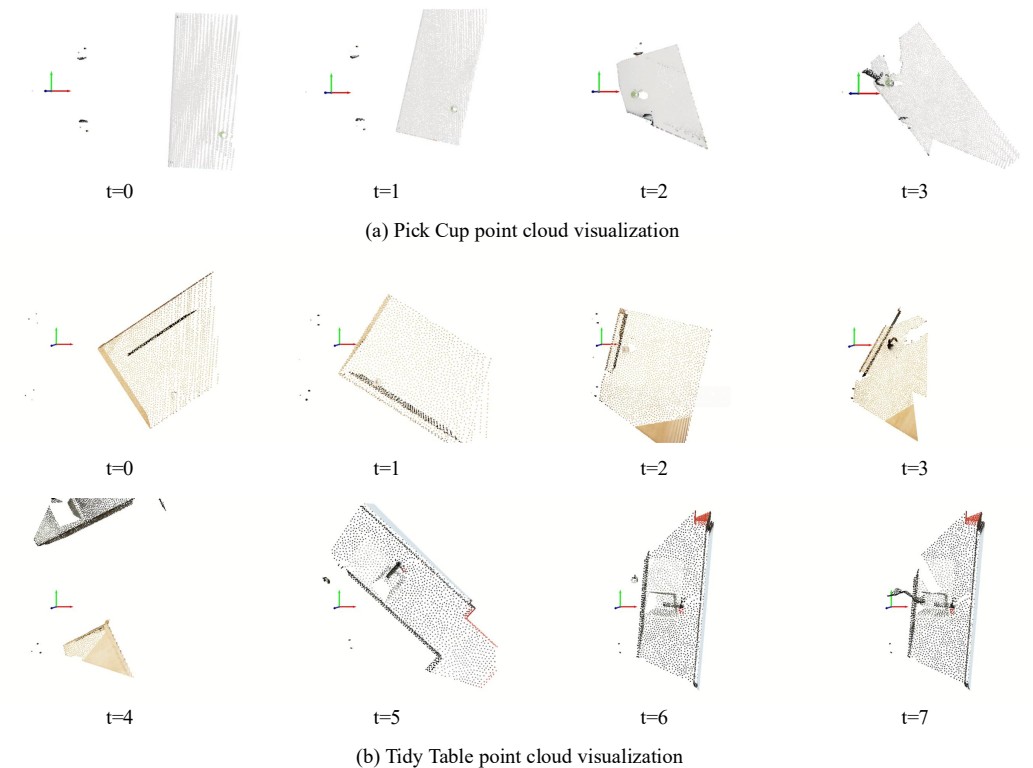

(a) Pick Cup point cloud visualization

(b) Tidy Table point cloud visualization

Figure 17: Visualization of ego-centric point cloud for Pick Cup and Tidy Table, used for WB-VIMA training. We fuse the colored point-cloud from three RGB-D cameras, crop with a robot-centric bounding box, and then downsample to 4096 points via farthest point sampling.

## C.2 WB-VIMA TRAINING DETAILS

**Policy Architecture.** WB-VIMA (Jiang et al., 2025a) takes as input both proprioceptive observations and an egocentric colored point cloud. The proprioceptive inputs include base velocity $v^{\text{base}} \in \mathbb{R}^3$, torso joints $q^{\text{torso}} \in \mathbb{R}^4$, left arm joints $q^{\text{left}} \in \mathbb{R}^6$, left gripper width $q^{\text{grip-left}} \in \mathbb{R}^1$, right arm joints $q^{\text{right}} \in \mathbb{R}^6$, and right gripper width $q^{\text{grip-right}} \in \mathbb{R}^1$. The point cloud is constructed by fusing RGB-D images from an eye-level camera and two wrist-mounted cameras; examples for Pick Cup and Tidy Table are shown in Figure 17. It is then cropped with a robot-centric bounding box and downsampled to 4096 points via farthest point sampling. WB-VIMA uses a 2-step history input. Additional architectural details are available in the original paper (Jiang et al., 2025a). For each task, we train WB-VIMA from scratch to obtain a single-task policy.

**Hyperparameters.** Training hyperparameters for the PointNet (point cloud processing), diffusion head, transformer backbone, learning rates, and task-specific point cloud clipping ranges are summarized in Table 4. The ego-centric clipping ranges are customized per task to reflect differences in scene layout and object randomization. All hyperparameters are chosen via grid search.

## C.3 $\pi_0$ TRAINING DETAILS

**Policy Architecture.** We fine-tune a pretrained $\pi_0$ model using LoRA with rank 32 for 50k steps and a batch size of 64. The model takes as input the RGB images from the eye-level and two wrist-mounted cameras, along with the same proprioceptive signals used by WB-VIMA, and predicts the target joint position for the next 50 time steps.

**Hyperparameters.** All RGB inputs are resized to $224 \times 224$. Actions and proprioceptive signals are normalized using the 1st–99th quantile and zero-padded to match the 32-dimensional action

| Hyperparameter | Value |
| --- | --- |
| Number of points in point cloud | 4096 |
| PointNet hidden dim | 256 |
| PointNet hidden depth | 2 |
| PointNet output dim | 256 |
| PointNet activation | GELU |
| Proprioceptive MLP input dim | 21 |
| Proprioceptive MLP hidden dim | 2 |
| Proprioceptive MLP hidden depth | 256 |
| Proprioceptive MLP output dim | 256 |
| Proprioceptive MLP activation | ReLU |
| Transformer embedding size | 512 |
| Transformer layers | 4 |
| Transformer heads | 8 |
| Transformer dropout rate | 0.1 |
| Transformer activation | GEGLU |
| Action dim | 21 |
| Unet down dims | [128, 256] |
| Unet kernel size | 5 |
| Unet number of groups | 8 |
| Diffusion step embedding dim | 256 |
| Diffusion noise scheduler | DDIM |
| Number of training steps | 100 |
| Beta schedule | squaredcos_cap_v2 |
| Number of denoise steps per inference | 16 |
| Learning rate | 1e-4 |
| Learning rate scheduler | Cosine decay |
| Learning rate warmup steps | 100000 |
| Learning rate cosine steps | 1300000 |
| Optimizer | AdamW |
| Batch size per GPU | 128 |
| Number of GPUs in parallel | 2 |
| Pick Cup (D0) pointcloud clip range | x: [0.0, 2.3], y: [-0.5, 0.5], z: [0.7, 2.0] |
| Pick Cup (D1) pointcloud clip range | x: [0.0, 2.7], y: [-1.0, 1.0], z: [0.7, 2.0] |
| Tidy Table (D0) pointcloud clip range | x: [0.0, 2.3], y: [-1.5, 1.5], z: [0.7, 1.5] |
| Model size | 37.1M |

Table 4: Hyperparameters for WB-VIMA.

| Hyperparameter | Value |
|---|---|
| Proprioceptive MLP input dim | 32 |
| Proprioceptive MLP output dim | 1024 |
| Flow Matching MLP input dim | 32 |
| Flow Matching MLP hidden dim | 2048 |
| Flow Matching MLP hidden depth | 2 |
| Flow Matching MLP output dim | 1024 |
| Flow Matching MLP activation | swish |
| PaliGemma embedding size | 2048 |
| PaliGemma number of layers | 18 |
| PaliGemma number of heads | 18 |
| PaliGemma heads dimension | 256 |
| PaliGemma MLP dimension | 16384 |
| Action expert embedding size | 1024 |
| Action expert MLP dimension | 4096 |
| Number of flow matching steps per inference | 10 |
| Learning rate | 2.5e-5 |
| Learning rate scheduler | Cosine decay |
| Learning rate warmup steps | 1000 |
| Learning rate cosine steps | 30000 |
| Optimizer | AdamW |
| Batch size | 64 |
| Number of GPUs in parallel | 4 |
| Model size | 3.3B |

Table 5: Hyperparameters for $\pi_0$.

space expected by $\pi_0$. The model uses a PaliGemma VLM backbone (Beyer et al., 2024) with a 300M-parameter action expert. Further architectural details are in (Black et al., 2024), and additional training hyperparameters are listed in Table 5.

## C.4 COMPUTE RESOURCES

With a batch size of 128, WB-VIMA can be trained on two RTX 3090 GPUs (24GB each), taking approximately 40 hours to reach 1 million steps. With a batch size of 64, $\pi_0$ can be trained on four H200 GPUs, taking approximately 7 hours to reach 50k steps.

## D USE OF LARGE LANGUAGE MODELS

We used a large language model (ChatGPT) solely for writing refinement, including grammar correction and improving clarity and conciseness of the text. The model was not used for research ideation, experimental design, analysis, or content generation beyond language editing.

