# OpenReview forum: "MoMaGen: Generating Demonstrations under Soft and Hard Constraints for Multi-Step Bimanual Mobile Manipulation"
_ICLR.cc/2026/Conference — ICLR 2026 Poster_

### Official Review · Reviewer_MKkL · 2025-10-27

**Soundness:** 2
**Presentation:** 3
**Contribution:** 3
**Rating:** 6
**Confidence:** 1

**Summary:**

The paper presents a demonstration generation framework through optimizing soft and hard constraints in bimanual mobile manipulation. Specifically, the hard constraints are formulated as the reachability, visibility during manipulation. The soft constraints are formed as the visibility while navigation. The proposed MoMAGen targets to satisfy the hard constraints while balance the soft constraints in the augmentation procedure of human demonstrations. The proposal is evaluated on four multi-step bimanual mobile manipulation tasks and it generates substantially more datasets than prior methods, which could be further exploited in the further imitation policy learning.

**Strengths:**

1.	The involving of reachability and visibility constraints are reasonable in the general data generation for multi-step bimanual mobile manipulation.
2.	Better diversity and task-relevant object visibility are achieved by MoMaGen compared to other baselines.
3.	The ablation studies are very comprehensive to investigate the influences of the two constraints in demonstration data generation.
4.	Greate policy learning is attained on the generated demonstrations by MoMaGen.
5.	The paper is clear written.

**Weaknesses:**

1.	One concern about this work is the efficiency of the demonstration collections and the computation cost of GPU for the constraint optimization.
2.	Is there any optimization for Eq.(1) when optimizing these constraints? Is there any conflicting terms in this equation? Some discussions should be included.

**Questions:**

please see the weaknesses.

---

> ### Author Response · Authors · 2025-11-21
> **Author Rebuttal**
>
> We would like to thank the reviewer for their constructive feedback. We appreciate that the reviewer acknowledges that our method produces “better diversity and task-relevant object visibility compared to other baselines”, the “ablation studies are very comprehensive”, and our paper is “clear written”. In the following paragraphs, we address the reviewer’s main questions and concerns, grouped by topic for clarity. **Notably, we added a new analysis of the computational cost of data generation (Section 5.5) in the revised manuscript.** The revised portions of our paper are highlighted in blue. If we have misunderstood any part of the reviewer’s question, we would be happy to further clarify and discuss during the discussion period.
>
> **Efficiency of Data Collection and Generation [Weakness 1]**
>
> Depending on task complexity, human teleoperation data collection takes 30 seconds to 5 minutes per episode. Additionally, subtask annotation takes 1 to 3 minutes per episode. Since our method only requires one source demonstration per task, the aforementioned time cost only needs to be paid once per task.
>
> Regarding the computational cost of data generation, each successful demonstration takes 0.1 to 1.3 GPU hours to generate (benchmarked on NVIDIA TITAN RTX GPUs). More details can be found in Appendix B.5. During our experimentation, human operators require approximately 30 seconds to 5 minutes per episode (including amortized setup and teardown), whereas our system requires 10-20x more wall-clock time per episode (measured on NVIDIA TITAN RTX GPUs). While slower per episode, our pipeline benefits from two major advantages: it can be parallelized across multiple GPUs, and it can run continuously without breaks. For example, a single human operator can realistically provide 30 hours of high-quality data per week (6 hours/day for 5 days). To generate the same volume of data, our pipeline would require 300-600 GPU-hours—achievable with only 2-4 TITAN RTX GPUs running continuously for a week. Thus, moderate GPU resources can match the optimistic weekly throughput of a human operator while eliminating fatigue, variance, and manual effort.
>
> To improve efficiency, we are developing a mechanism that rewinds to a recent valid state and retries only the failed subtask, rather than restarting from scratch. We expect this to significantly boost throughput and reduce compute cost for complex tasks where data generation success rates are low, and we plan to include this improvement in our upcoming open-source release.
>
> **Optimization of Equation 1 and Discussion about Conflicting Terms [Weakness 2]**
>
> We accelerate data generation by prioritizing fast IK checks over full motion planning
> whenever possible for preemptive filtering, and by decomposing robot’s configuration into torso/arm subspaces for efficient conditional sampling. Feel free to check out Section 4.2 and Algorithm 1 as a concrete instantiation and efficient implementation of Equation 1.
>
> Regarding the “conflicting terms” in Equation 1, it is often the case that the objective function (which aggregates soft constraints) conflicts with the hard constraints. For instance, a common soft constraint in motion planning is minimizing trajectory length to reduce unnecessary motion. This can conflict with the task-success hard constraint: remaining stationary minimizes trajectory length but clearly does not complete the task. More broadly, the hard constraints define the feasible set and guide the algorithm toward trajectories that are not only successful and collision-free but also smooth and efficient.

---

> ### Comment · Reviewer_MKkL · 2025-11-26
> **Concerns have been addressed**
>
> Thanks for the authors feedback. My concerns about the training efficiency and the optimization conflicts have been addressed. Good analysis is presented on conflicting terms and I will keep my initial rating.

---

> > ### Author Response · Authors · 2025-11-26
> > **Author Response**
> >
> > Dear Reviewer MKkL,
> >
> > Thank you again for your thoughtful feedback. If you feel that our rebuttal has adequately addressed your concerns, we are wondering whether you could consider updating your rating accordingly. If not, we completely understand and sincerely appreciate the time you’ve devoted to our submission.
> >
> > Best regards,
> >
> > Authors

---

> ### Author Response · Authors · 2025-11-26
> **Thank you**
>
> Dear MKkL,
>
> Thank you very much for your prompt response. We really appreciate your review and feedback!
>
> Best,
>
> Authors

---

### Official Review · Reviewer_Nrsb · 2025-10-31

**Soundness:** 3
**Presentation:** 3
**Contribution:** 3
**Rating:** 6
**Confidence:** 4

**Summary:**

This manuscript introduces MoMaGen, a novel framework designed to automate the generation of large-scale, diverse demonstrations for complex multi-step bimanual mobile manipulation tasks. The work addresses the significant challenge and cost associated with collecting such high-dimensional data via traditional teleoperation, which requires controlling a mobile base, dual high-DoF arms, and active vision simultaneously. MoMaGen leverages a constraint-based system to generate these demonstrations in simulation efficiently. The research validates the quality of the resulting comprehensive dataset by successfully training and testing two SOTA imitation learning baselines.

**Strengths:**

1. **Pioneering Problem Scope**: This research is groundbreaking as it is one of the first to focus on the full-body control problem, which integrates a mobile base, active vision, and dual-arm collaborative manipulation. This novel scope holds significant, pioneering implications for general-purpose robot control.
2. **Comprehensive Dataset Contribution**: The study constructs the most comprehensive MoMaGen dataset to date within the X-Gen series. This large-scale, diverse resource is invaluable for advancing the research and development of mobile manipulation learning.
3. **Superior Data Acquisition Efficiency**: The MoMaGen framework demonstrates an effective method for acquiring high-quality demonstration data with relatively lower cost and higher efficiency compared to mainstream, traditional teleoperation methods, contributing a smart solution to the data bottleneck problem.
4. **Thorough Empirical Validation**: The authors performed substantial work by rigorously testing two SOTA imitation learning baselines on the newly generated MoMaGen dataset, which effectively substantiates the quality and utility of the synthesized data for policy learning.

**Weaknesses:**

1. **Over-reliance on Heuristics for Automation**: Despite the formal definition of numerous hard and soft constraints, the actual demonstration generation process in simulation still heavily relies on manual intervention, such as simple inverse kinematics, heuristic rules, and human-provided one-shot annotations.

2. **Limited Task Generalization and Scalability**: The study currently only covers a small number of distinct task types (a total of four). This limited diversity is significantly less than what is typically observed in large-scale, simulator-based demonstration generation work, raising concerns about MoMaGen's efficiency and ability to scale up for generating true variety in large datasets.

3. **Unconvincing Real-Robot Validation**: The physical robot experiments are narrow in scope, only covering the simplest 'pick cup' task with minimal cross-embodiment transfer and a scene closely resembling the simulation. The low final success rate, even after fine-tuning with additional dozens of real-world demonstrations, suggests that the proposed system's real-world robustness is limited, and simpler baselines (like ACT or DP) might yield similar performance with comparable fine-tuning.

**Questions:**

1. **Automation Gap**: How can the reliance on manual annotations and heuristic checks for constraints (e.g., visibility, reachability) be eliminated to achieve a fully automated demonstration generation pipeline?

2. **Scaling Task Diversity**: What specific, proposed architectural or procedural updates will be implemented to increase the framework's capability to generate a wide, scalable range of tasks beyond the current four?

3. **Real-World Generalization**: What concrete steps will be taken to validate the system on more complex, multi-step tasks in the real world, and to demonstrate improved generalization or cross-embodiment transfer with less fine-tuning data?

---

> ### Author Response · Authors · 2025-11-21
> **Author Rebuttal Part 1**
>
> We thank the reviewer for recognizing our work as pioneering in full-body control, for noting the breadth of our dataset relative to prior efforts, and for highlighting both the high quality and low cost of our generated data as well as the thoroughness of our empirical evaluations. In the following paragraphs, we address the reviewer’s main questions and concerns, grouped by topic for clarity. **Notably, we added three additional experiments that demonstrate MoMaGen’s capabilities beyond pick-and-place, covering (1) articulated object manipulation, (2) pouring (particle transfer), and (3) whole-body control (Appendix B.1) in the revised manuscript. The videos of the generated demos for these new tasks are shown under Task Diversity on our project website at https://momagen-iclr2026.github.io/.** The revised portions of our paper are highlighted in blue. If we have misunderstood any part of the reviewer’s question, we would be happy to further clarify and discuss during the discussion period.
>
> **Over-reliance on Heuristics and Manual Intervention [Weakness 1, Question 1]**
>
> We would like to clarify that the only manual intervention needed for MoMaGen is the subtask annotation. Each annotation takes only 1 to 3 minutes and is done once per task, resulting in minimal overhead. All other components, including the full constraint-optimization pipeline, are fully automated, covering: base and head pose sampling within heuristic ranges, collision checking in simulation, inverse-kinematics reachability checks, motion planning for feasibility, and trajectory execution using built-in joint- and task-space controllers.
>
> Regarding the reviewer’s suggestion of full automation, we have explored automated subtask annotation in preliminary experiments. For contact-based segmentation, privileged simulation signals like contact forces or collisions could help identify transitions. For visual-based segmentation, video understanding models and VLMs may learn subtask boundaries directly from demonstrations. While promising, both approaches can be error-prone in the presence of noise or suboptimal behavior. Since our method relies on just one source demo per task, we chose manual annotation to ensure the highest quality possible. Nonetheless, we consider automating this step a worthwhile future direction for enhancing scalability, should multiple source demonstrations prove useful.
>
> **Limited Task Generalization and Scalability & Scaling Task Diversity [Weakness 2, Question 2]**
>
> We appreciate the reviewer’s question regarding MoMaGen’s ability to scale to a broader set of tasks.
>
> In the original manuscript, our four core tasks already demonstrated that MoMaGen can handle common families of bimanual mobile manipulation tasks involving coordinated, contact-rich, and multi-step motions.
>
> During the rebuttal period, we extended our evaluation by adding three new experiments that further illustrate MoMaGen’s generality beyond pick-and-place and contact-rich manipulation. These new tasks span articulated-object manipulation, particle-based pouring, and coordinated whole-body control:
>
> 1. Get Bottle from Fridge (Articulated Object Manipulation). The robot navigates to a fridge, grasps and opens the door with one arm, and retrieves a bottle from inside.
> 2. Pour Cat Food (Pouring and Particle Interaction).
> The robot grasps a tin containing rigid particles, navigates to a bowl, and performs a controlled pouring motion until all particles are transferred.
> 3. Push Chair (Whole-Body Control).
> The robot navigates to the back of a chair, establishes contact, and pushes it into position under a table.
>
> MoMaGen successfully generated demonstrations for all three tasks without any modification to the pipeline, highlighting its flexibility and generality across diverse and challenging bimanual mobile manipulation scenarios.
>
> Further details are provided in the updated Appendix B.1, and videos of the generated demonstrations are available under Task Diversity on our project website at https://momagen-iclr2026.github.io/.

---

> ### Author Response · Authors · 2025-11-21
> **Author Rebuttal Part 2**
>
> **Unconvincing Real-Robot Validation & Real-World Generalization [Weakness 3, Question 3]**
>
> We appreciate the reviewer’s concern regarding the scope of our real-world experiments. It is correct that our physical validation focuses on a single task (Pick Cup) and does not include more complex, coordinated tasks such as Clean Frying Pan. We selected Pick Cup as an initial real-world testbed to specifically evaluate sim-to-real transfer of MoMaGen-generated data, while avoiding confounding failure modes that arise in more complex bimanual coordination scenarios.
> While simpler than Clean Frying Pan, Pick Cup still requires precise mobile base placement to ensure the cup is within the robot’s reachable workspace, an essential challenge in mobile manipulation. In fact, many observed sim-to-real failures stem from the base stopping slightly too close or too far from the table. Thus, even this “simple” task provides meaningful insight: MoMaGen’s diverse synthetic demonstrations offer a strong prior that enables effective policy learning in low-data real-world regimes.
> We fully agree that validating more complex tasks, including bimanual coordination, is an important next step. Unfortunately, our Galaxea R1 robot is currently under repair and will not be operational during the rebuttal period. Once the hardware is restored, we intend to experiment on more complex tasks as part of our continued work on this research agenda.
> Regarding the final real-world success rate, we emphasize that achieving 60% success with only 40 real-world demonstrations for fine-tuning is non-trivial. State-of-the-art VLA models such as pi05/06 experience substantial performance degradation in mobile manipulation settings, largely due to the difficulty of collecting large-scale mobile manipulation datasets, a gap MoMaGen is designed to help mitigate. Real-world performance could potentially be improved with additional visual randomization in simulation or a real-to-sim-to-real pipeline, which is beyond the scope of this work.
> Regarding baseline choice, we did not include ACT or DP because WB-VIMA significantly outperforms both methods in the mobile manipulation domain, as shown in [1]. Given that WB-VIMA achieves 10% success with 40 demos, we expect ACT and DP to perform worse, offering limited additional insight.
> Finally, regarding cross-embodiment transfer, Appendix B.2 shows that a demonstration collected on the Galaxea R1 can be used by MoMaGen to generate trajectories for a TIAGo robot. Although this works to some extent, since key task-space poses transfer across embodiments, the data generation success rate is naturally lower due to differences in kinematics. A systematic study of sim-to-real cross-embodiment transfer would therefore require collecting and generating demonstrations across multiple robot embodiments (analogous to pi0’s multi-embodiment pretraining), followed by fine-tuning on a new robot. While this is outside the scope of the current paper, we see multi-embodiment synthetic pretraining as an exciting and impactful direction for future work.
>
> [1] Jiang, Yunfan, et al. "Behavior robot suite: Streamlining real-world whole-body manipulation for everyday household activities." arXiv preprint arXiv:2503.05652 (2025).

---

### Official Review · Reviewer_8M33 · 2025-11-01

**Soundness:** 3
**Presentation:** 3
**Contribution:** 3
**Rating:** 6
**Confidence:** 3

**Summary:**

The paper proposes MoMaGen, a data generation framework which leverages soft (object visibility during navigation, retraction) and hard constraints (visibility during manipulation, reachability) to generate mobile manipulation demonstrations (with navigation and manipulation in separate stages). The authors evaluate the data generated by MoMaGen (diversity, object visibility, cross-embodiment support), policies learned with MoMaGen demonstrations (SR compared with baselines, object visibility, data scaling), and sim2real via pretraining on MoMaGen demonstrations and finetuning on real demos.

**Strengths:**

- Hard and soft visibility constraints are novel, and per experiments improve performance across tasks
- Method is validated on a range of free-space and contact-rich tasks, at varying levels of randomization
- Thorough experiments on the generated data and policy learning with MoMaGen data
- Method works well with just one human demonstration, reducing human supervision requirements
- Real-world transfer experiment (finetuned with 40 real demos), demonstrates benefits of pretraining on MoMaGen's generated demonstrations

**Weaknesses:**

- The current setup does not provide demonstrations for coordinated upper and lower-body control, a key developing area in mobile manipulation research
- Authors note that each successful demonstration takes ~0.1-1.3 GPU hours, which can substantially limit large-scale data generation. Such large-scale generation is important for especially complex, long-horizon mobile manipulation tasks

**Questions:**

- What are the major causes for the relatively large computational cost per successful demonstration, i.e. is this due to the computational costs of the optimization scheme, due to the underlying simulator, etc? Are there means to reduce the computational cost to generate data more quickly, especially when given access to fewer GPUs?
- Are there empirical benefits in MoMaGen to providing additional teleoperated demonstrations (since, once a teleoperation setup is built, generating and annotating a few source demos is approximately as difficult as generating a single demo)?

Small notes:
- On page 6, final paragraph, "robot-specific kinematic" should be "robot-specific kinematics"
- On page 19, in the Fig 13 caption, "downsampe" should be "downsample"

---

> ### Author Response · Authors · 2025-11-21
> **Author Rebuttal**
>
> We thank the reviewer for recognizing the novelty of our proposed soft and hard visibility constraints, the validity of our method across diverse tasks, the thoroughness of our experiments, and the benefits demonstrated in our real-world evaluations using MoMaGen-generated data. In the following paragraphs, we address the reviewer’s main questions and concerns, grouped by topic for clarity. **Notably, we 1) added an additional experiment that demonstrated MoMaGen’s whole-body control capability on a new Push Chair task (Appendix B.1; also shown under Task Diversity on our project website at https://momagen-iclr2026.github.io/), and (2) added a new analysis of the computational cost of data generation (Section 5.5) in the revised manuscript.** The revised portions of our paper are highlighted in blue. If we have misunderstood any part of the reviewer’s question, we would be happy to further clarify and discuss during the discussion period.
>
> **Limited coordination between upper-body and lower-body control [Weakness 1]**
>
> We appreciate the reviewer’s insight and agree that whole-body manipulation requiring tight coordination between the upper and lower body is an important aspect of mobile manipulation.
>
> To address this concern, we conducted an additional experiment during the rebuttal period. We used MoMaGen to generate demonstrations for a new Push Chair task that explicitly requires coordinated whole-body control: the robot must use both arms and its mobile base to push a chair and maneuver it under a table. This task inherently demands tight coupling between arm and base motions.
>
> MoMaGen successfully generated new demonstrations for this setting. Concretely, we (1) invoked the base motion planner to reach the base pose of the source demonstration, (2) invoked the arm motion planner to reach the corresponding end-effector poses, and (3) replayed the whole-body task-space control sequence to achieve task success. A video of a generated demonstration is provided on our project website under “Task Diversity” at https://momagen-iclr2026.github.io/.
>
> **Computation cost analysis and mitigation for data generation [Weakness 2, Question 1]**
>
> Thank you for this insightful question. To answer this, we have included a detailed computational cost analysis of the key components in our data generation pipeline. Specifically, we measure the wall-clock time spent in base sampling, motion planning, and simulation execution. We find that simulation execution dominates total compute time, often exceeding planning durations by a large margin – for example, executing a planned base motion in simulation takes 100 seconds on average, compared to 18 seconds for planning. We also observe substantial variance in base sampling time due to our current random sampling strategy, which becomes especially costly when feasible base poses are limited. Furthermore, we report the average compute cost of each component as a percentage of total episode duration (computed only over successful episodes for fairness), revealing that the cost of base sampling increases significantly from D0 to D2 as scene complexity rises. Please refer to Section 5.5 for more details and the corresponding figures.
>
> To improve efficiency, we are developing a mechanism that rewinds to a recent valid state and retries only the failed subtask, rather than restarting from scratch. We expect this to significantly boost throughput and reduce compute cost for complex tasks where data generation success rates are low, and we plan to include this improvement in our upcoming open-source release.
>
> **Benefits of using more than one source demonstration [Question 2]**
>
> We intentionally use one source demonstration per task to minimize data collection and annotation effort, which we consider a key strength of our approach. That said, collecting a small number of additional demonstrations (e.g., 5-10) is certainly feasible. Prior work in the X-Gen series has explored simple heuristics for choosing the “closest” source demonstration to emulate from a pool, but such heuristics do not directly transfer to the mobile-manipulation setting due to the added complexity of base placement and whole-body coordination.
> We agree that using multiple source demonstrations could offer benefits. For example, demonstrations that vary in subtask ordering or execution details (e.g., grasp type or approach direction) may allow the system to choose motions that better match the current scene configuration. We view this as a promising avenue for future work that could balance human effort with improved generalization from a richer pool of source demonstrations.
>
> **Typos**
>
> We have corrected all identified typos in the revised manuscript. Thank you for pointing them out!

---

> > ### Comment · Reviewer_8M33 · 2025-11-26
> > **Response to authors**
> >
> > Thank you to the authors for their responses; most of my questions have been answered. However, the example of the push chair task seems to alternate between base and arm movements, and it does not seem MoMaGen supports simultaneous base and arm movements. Can the authors confirm this point?

---

> ### Author Response · Authors · 2025-11-26
> **Author Response 1**
>
> Dear Reviewer 8M33,
>
> Thank you very much for your prompt response. MoMaGen does support simultaneous base and arm movements. In fact, our motion planner supports setting both base pose and end-effector poses as goals and can plan a trajectory of simultaneous base and arm motion to reach those goals. Once the robot arrives at the pre-contact poses, it can replay the base and end-effector motion from the source demo simultaneously as well, for the contact-rich portion of the task. We hope this clarifies and please let us know if you have any follow-up questions!
>
> Best,
>
> Authors

---

> > ### Comment · Reviewer_8M33 · 2025-11-26
> > **Response to authors**
> >
> > I thank the authors for their additional clarifications, and the additional computational cost analysis and push chair experiments address my concerns. Trajectory generation speed remains a weakness, even removing simulation rollout time. However, I believe the remainder of the work is well-presented and provides novel contribution, hence I have increased my score to an 8.

---

> > > ### Author Response · Authors · 2025-11-26
> > >
> > > Dear Reviewer 8M33,
> > >
> > > Thanks a lot for your review and feedback!
> > >
> > > Best,
> > >
> > > Authors

---

### Official Review · Reviewer_scaJ · 2025-11-01

**Soundness:** 3
**Presentation:** 4
**Contribution:** 3
**Rating:** 6
**Confidence:** 5

**Summary:**

MoMaGen addresses the challenge of generating diverse demonstration data for bimanual mobile manipulation tasks, where collecting human teleoperation data is prohibitively expensive due to the complexity of controlling both the mobile base and two arms simultaneously. The key contribution is formulating data generation as a constrained optimization problem that satisfies hard constraints (reachability, collision avoidance, object visibility during manipulation) while optimizing soft constraints (visibility during navigation, compact retraction). Unlike prior methods (MimicGen, SkillMimicGen, DexMimicGen) that fail on mobile manipulation, MoMaGen jointly optimizes base pose, camera pose, and end-effector trajectories to generate diverse, high-quality demonstrations from a single human demonstration. Evaluated on four household tasks with aggressive randomization, MoMaGen achieves 63% generation success rate and 75-100% object visibility (vs 35-65% for baselines), enabling policies trained on synthetic data to outperform baselines and successfully transfer to real hardware with minimal fine-tuning (60% vs 0% success with π₀ using 40 real demos).

**Strengths:**

**Originality**

The paper makes several original contributions. The constrained optimization formulation elegantly unifies existing X-Gen methods while introducing novel visibility and reachability constraints specific to mobile manipulation. The distinction between hard constraints (must satisfy) and soft constraints (desirable) is intuitive and principled. Notably, the work is the first to tackle automated data generation for bimanual mobile manipulation, addressing visibility of moving cameras and reachability with mobile bases—problems unexplored in prior work. However, the core technical components (motion planning, IK solving, task-space replay) largely build on existing tools (cuRobo), with the main innovation being their orchestration under the constraint framework. The cross-embodiment generation, while interesting, is acknowledged to have significant limitations (gripper size, workspace differences) that limit its practical applicability.

**Quality**

The experimental methodology is rigorous and comprehensive. The three-level randomization scheme systematically evaluates increasing difficulty, and comparisons against two strong baselines are fair. The evaluation covers multiple important dimensions: data diversity metrics, generation success rates, visibility analysis, and downstream policy performance with two different algorithms. Ablation studies effectively demonstrate the contributions of hard and soft visibility constraints. The sim-to-real experiments, while limited to one simple task, provide important validation.

**Clarity**

The paper is well-structured with clear mathematical formulation and effective visualizations. Some technical details lack precision—the "heuristics" for base pose sampling (line 8, Algorithm 1) are mentioned but not specified, making full reproducibility difficult despite code availability claims.

**Significance**

This work addresses a critical bottleneck in scalable robot learning for household tasks, and the ability to generate diverse demonstrations from a single source is valuable. The constrained optimization framework provides a principled foundation for future methods. The sim-to-real transfer demonstrates practical utility: achieving 60% success with $\pi_{0}$ using only 40 real demonstrations (versus 0% without pretraining) shows that synthetic data provides a strong prior that improves sample efficiency in the real world. This represents meaningful and well-validated progress toward scalable mobile manipulation learning with clear pathways for real-world deployment.

**Weaknesses:**

**Limited Real-World Validation and Inadequate Scaling Analysis**

The sim-to-real evaluation is insufficiently comprehensive. Only the simplest task (Pick Cup) is deployed on real hardware, achieving modest success rates (10% for WB-VIMA, 60% for π₀), which fails to validate whether the diversity benefits generalize to multi-step bimanual coordination tasks. Furthermore, while Figure 7 demonstrates data scaling trends in simulation, the choice of specific quantities (500, 1000, 2000 demonstrations) is not justified, and no corresponding analysis exists for real-world deployment. The paper does not explain why 1000 demonstrations is chosen as the primary experimental setting, nor does it investigate whether this quantity represents an optimal trade-off between generation cost and policy performance. The optimal quantity of synthetic data for sim-to-real transfer remains unexplored. A rigorous evaluation should include at least one complex multi-step task on physical hardware, provide principled justification for dataset sizes, and conduct systematic scaling experiments to characterize sample efficiency in real-world settings.

**Unquantified Annotation Overhead**

While the method claims to require "only a single source demo," the annotation requirements detailed in Section 4.2 (target object, gripper-held object, pre-grasp timestep, end timestep, retraction type) represent non-trivial human effort that is neither quantified nor analyzed. The absence of timing measurements or discussion of automation strategies (e.g., leveraging vision-language models for object identification or contact detection for grasp inference) limits assessment of the method's practical scalability. A complete evaluation should quantify annotation time per task and investigate semi-automated annotation approaches.

**Underspecified Sampling Methodology**

The base pose sampling procedure (Algorithm 1, line 8) lacks sufficient mathematical specification. The paper references "heuristics" and sampling "near target objects" without defining the probability distribution, spatial bounds, rejection criteria, or termination conditions. This ambiguity impedes reproducibility and precludes theoretical analysis of convergence properties or sample complexity. The method requires rigorous formalization including explicit distributions, parameterized bounds, and well-defined stopping criteria.

**Absence of Failure Mode Characterization**

Despite generation success rates ranging from 22-63% across randomization levels, the paper provides no taxonomic analysis of failure modes. The relative contributions of base sampling failures, inverse kinematics infeasibility, motion planning failures, and simulation instabilities remain uncharacterized. This lack of diagnostic information obscures the primary algorithmic bottlenecks and limits targeted improvements. A systematic failure analysis categorizing failure types by frequency and correlating them with scene complexity, task structure, and randomization level would provide essential insights for future development.

**Insufficient Computational Cost Analysis**

The reported computational cost (0.1-1.3 GPU hours per demonstration, Appendix B.5) lacks contextualization and granular analysis. The paper does not decompose this cost across algorithmic components (motion planning, inverse kinematics, sampling iterations, simulation), nor does it provide comparative analysis against human teleoperation baselines. Without quantitative cost-benefit analysis comparing alternative data collection strategies (e.g., 1000 human demonstrations versus 1 human + 999 synthetic), practitioners cannot make informed deployment decisions. A rigorous efficiency analysis should profile computational bottlenecks and establish cost-effectiveness relative to conventional data acquisition methods.

**Questions:**

**Questions**

**Q1: Base Pose Sampling Specification** - What is the exact probability distribution for sampling base poses (Algorithm 1, line 8)? What are the spatial bounds relative to target objects? How many sampling attempts occur before declaring failure? This specification is critical for reproducibility and theoretical analysis of the method's convergence properties.

**Q2: Failure Mode Distribution** - What percentage of generation failures are attributable to base sampling timeout, IK infeasibility, motion planning failures, and simulation instabilities respectively? Given that 37-78% of attempts fail across randomization levels, understanding the primary bottleneck is essential for targeted improvements.

**Q3: Real-World Scaling and Task Complexity** - Why were multi-step bimanual tasks (Tidy Table, Clean Frying Pan) not evaluated on real hardware? Does the sim-to-real gap remain consistent across task complexity, or do coordination-heavy tasks exhibit degraded transfer?

**Suggestions**

**S1: Formalize Base Sampling Methodology** - Provide rigorous mathematical specification of the base pose sampling procedure in the main paper, including explicit probability distributions, parameterized spatial bounds, and well-defined termination criteria. This is essential for reproducibility and enables theoretical analysis.

**S2: Comprehensive Failure Analysis** - Include a detailed breakdown categorizing failure types by frequency across tasks and randomization levels. Correlate failures with scene characteristics (object density, workspace constraints) to identify systematic limitations and guide algorithmic improvements.

---

> ### Author Response · Authors · 2025-11-21
> **Author Rebuttal Part 1**
>
> We would like to thank the reviewer for their constructive feedback. We appreciate that the reviewer acknowledges that our constrained optimization formulation elegantly unifies existing X-Gen methods, our method is the first to tackle automated data generation for bimanual mobile manipulation, the experiments are rigorous and comprehensive, and our paper addresses a critical bottleneck in scalable robot learning for household tasks. In the following paragraphs, we address the reviewer’s main questions and concerns, grouped by topic for clarity. **Notably, we added additional analysis for 1) data generation failure mode and 2) computational cost (in Section 5.5 and Appendix B.6) in the revised manuscript.** The revised portions of our paper are highlighted in blue. If we have misunderstood any part of the reviewer’s question, we would be happy to further clarify and discuss during the discussion period.
>
> **Limited Real-World Validation [Weakness 1 (a) and Question 3]**
>
> We appreciate the reviewer’s concern regarding the scope of our real-world experiments. It is true that our physical validation was conducted on one task only (Pick Cup) and did not include bimanual manipulation. We chose this task as an initial testbed for real-world deployment to focus on validating the sim-to-real transferability of MoMaGen-generated data and to isolate the impact of our data generation pipeline without introducing compounding failure modes from complex coordination.
>
> Note that even though Pick Cup is simpler than other tasks like Clean Frying Pan, it still requires accurate robot base placement so that the cup is within the reachable workspace and amenable for grasping, an important challenge in mobile manipulation. In fact, many of our sim-to-real failures result from the robot base being either too close or too far away from the table. Despite its apparent simplicity, the Pick Cup sim-to-real experiment provides meaningful insight: diverse synthetic data by MoMaGen provides a valuable prior for efficient policy learning in low-data regimes of the real-world.
>
> With that being said, we very much agree with the reviewer that validating more complex tasks, including those involving bimanual coordination, is an important next step. We also agree with the reviewer that exploring the optimal quantity of synthetic data for sim-to-real transfer (trade-off between generation cost and policy performance) is an exciting research direction. An immediate next step we can do in this regard is to conduct the same sim-to-real transfer with the checkpoints trained with 500 and 2000 synthetic demonstrations. Unfortunately, our robot (Galaxea R1) is currently out of service and under repair, and it is unlikely to be functional within the rebuttal period. Once the hardware is restored, we intend to carry out these additional scaling-analysis experiments and incorporate more complex tasks as part of our continued work on this research agenda.
>
> **Inadequate Scaling Analysis [Weakness 1 (b)]**
>
> We appreciate the reviewer’s request for a principled justification of our dataset sizes. We choose 1000 demonstrations as our primary experimental setting to follow the conventions established by prior works (MimicGen, SkillMimicGen, DexMimicGen), which serve as our baselines and allow for a fair comparison. These works show that even for relatively simple tabletop manipulation tasks, policy performance typically saturates only when using around 1000 demonstrations. Given that our multi-step mobile manipulation tasks are inherently more complex, we consider 1000 demonstrations to be a reasonable starting point. To study scaling behavior, we additionally evaluate 500 and 2000 demonstrations (0.5x and 2x the default), again following standard practice.
>
> **Unquantified Annotation Overhead [Weakness 2]**
>
> We acknowledge that our method requires annotating contact mode change timesteps for each manipulated object, which might seem tedious. In reality, each annotation takes only 1 to 3 minutes and is done once per task, resulting in minimal overhead.
>
> We appreciate the reviewer’s suggestion to automate subtask annotations. For contact-based segmentation, privileged simulation signals like contact forces or collisions could help identify transitions. For visual-based segmentation, video understanding models and VLMs may learn subtask boundaries directly from demonstrations.
>
> While promising, both approaches can be error-prone in the presence of noise or suboptimal behavior. Since our method relies on just one source demo per task, we chose manual annotation to ensure the highest quality possible. Nonetheless, we consider automation a worthwhile future direction for enhancing scalability, should multiple source demonstrations prove useful.

---

> ### Author Response · Authors · 2025-11-21
> **Author Rebuttal Part 2**
>
> **Underspecified Sampling Methodology [Weakness 3, Question 1, Suggestion 1]**
>
> We adopt a simple heuristic for robot base sampling (Algorithm 1, line 8). Specifically, MoMaGen samples a random distance between MIN_DIST (0.4 m) and MAX_DIST (1.0 m), and a random angle between -pi and pi. This corresponds to sampling a random 2D point in a ring-shaped region centered at the target object, where the inner radius is MIN_DIST and the outer radius is MAX_DIST. The robot is always oriented to face the target object. The values of MIN_DIST and MAX_DIST can be tuned for different robots depending on their arm reach and kinematic specifications. If a sampled base pose results in a collision, it is immediately discarded and re-sampled.
>
> Similarly, to sample an eye pose (Algorithm 1, line 9), we first sample a random eye position by adding a random offset to the default eye pose: 0 to 0.3m in x, -0.05 to 0.05m in y, -0.2 to 0.1m in z. We then sample a random eye orientation from a cone centered on the ray from the sampled eye position to the target object, using a half-angle of 30 degrees. All of these ranges are hyperparameters that can be tuned for different robot embodiments and to balance data generation success rate versus data diversity. We have included the data generation source code in the supplementary material for full reproducibility.
>
> **Absence of Failure Mode Characterization [Weakness 4, Question 2, Suggestion 2]**
>
> Thank you for this valuable suggestion. In response, we now provide a detailed characterization of failure modes observed during data generation. Unsuccessful episodes are categorized into base sampling, base-level planning, arm-level planning, and simulation instabilities. We find that simulation instabilities (e.g., controller inaccuracies and stochastic effects in the simulator) account for a substantial portion of failures (35%). Among planner-related issues, arm-level planning dominates (40% on average), while base-level planning accounts for 26%. Additionally, in D2 randomization, navigation-related failures (base sampling, base IK, base TrajOpt) increase significantly due to floor obstacles in an already constrained space. We also include a step-wise breakdown of failures across multi-step tasks (Fig. 16 in Appendix). Please refer to Section 5.5 and Appendix B.6 for full details and supporting figures.
>
> **Insufficient Computational Cost Analysis [Weakness 5]**
>
> Thank you for this valuable suggestion. In response, we now include a detailed computational cost analysis of the key components in our data generation pipeline. Specifically, we measure the wall-clock time spent in base sampling, motion planning, and simulation execution. We find that simulation execution dominates total compute time, often exceeding planning durations by a large margin – for example, executing a planned base motion in simulation takes 100 seconds on average, compared to 18 seconds for planning. We also observe substantial variance in base sampling time due to our current random sampling strategy, which becomes especially costly when feasible base poses are limited. Furthermore, we report the average compute cost of each component as a percentage of total episode duration (computed only over successful episodes for fairness), revealing that the cost of base sampling increases significantly from D0 to D2 as scene complexity rises. Please refer to Section 5.5 for more details and the corresponding figures.
>
> We also compare our automated data generation pipeline to human teleoperation. Human operators require approximately 30 seconds to 5 minutes per episode (including amortized setup and teardown), whereas our system requires 10-20x more wall-clock time per episode (measured on NVIDIA TITAN RTX GPUs). While slower per episode, our pipeline benefits from two major advantages: it can be parallelized across multiple GPUs, and it can run continuously without breaks. For example, a single human operator can realistically provide 30 hours of high-quality data per week (6 hours/day for 5 days). To generate the same volume of data, our pipeline would require 300-600 GPU-hours—achievable with only 2-4 TITAN RTX GPUs running continuously for a week. Thus, moderate GPU resources can match the optimistic weekly throughput of a human operator while eliminating fatigue, variance, and manual effort.
>
> To improve efficiency, we are developing a mechanism that rewinds to a recent valid state and retries only the failed subtask, rather than restarting from scratch. We expect this to significantly boost throughput and reduce compute cost for complex tasks where data generation success rates are low, and we plan to include this improvement in our upcoming open-source release.

---

### Author Response · Authors · 2025-12-03
**Final Remarks by Authors to the AC**

Dear AC,

Thank you very much for your attention to our paper. We believe our rebuttal has comprehensively addressed the reviewers' concerns through substantial new experiments, expanded analyses, and clarifications, resolving key misunderstandings. Below, we summarize the major additions made during the rebuttal period.

**Failure mode characterization:** We added a detailed characterization of failure modes in the data generation process, specifying both the types of failures and the stage of the pipeline at which each occurs. These additions appear in Section 5.5 and Appendix B.6.

**Computational cost analysis:** We provided a fine-grained computational cost analysis of the key components in our data generation pipeline. The analysis is presented in Section 5.5.

**Expanded Task Diversity (three new tasks):** To address concerns regarding limited task diversity, we added three additional experiments that demonstrate MoMaGen's capabilities beyond pick-and-place:
1. Get Bottle from Fridge (Articulated Object Manipulation)
2. Pour Cat Food (Pouring and Particle Interaction)
3. Push Chair (Whole-Body Control)

All corresponding results appear in Appendix B.1, and videos of the newly generated demonstrations are available under Task Diversity on our project website: https://momagen-iclr2026.github.io/.

We have also addressed all reviewer comments and clarified all outstanding questions. We appreciate that Reviewer 8M33 noted the strength of our rebuttal and increased their rating accordingly.

We kindly request that the AC consider these clarifications and newly added results when making the final decision. Thank you for your time and consideration.

Best,

Authors

---

### Meta-Review · Area_Chair_wKhb · 2026-01-06

**Summary:**

The paper studies data generation for Multi-Step Bimanual Mobile Manipulation. All reviewers are positive about the work, noting that the problem is novel and the approach is rigorous and well-evaluated. The authors effectively addressed the reviewers' concerns, adding more information about failure modes and new tasks. Overall, it is a solid paper, and the AC recommends acceptance.

The authors promise more real-world validation beyond pick-and-place once the robot is repaired. The AC hopes the authors will include this in the final version.

**Reviewer Concerns:**

Concerns addressed:

- Failure modes and analysis
- Greater tasks diversity
- Computational cost analysis

Outstanding concern:
- More real world validaton beyond pick and place.

**Reviewer Scores:**

All reviewers are positive. A few might have increased their score after the rebuttal.

---

### Decision · Program_Chairs · 2026-01-26

Accept (Poster)